# Immune Regulation, but Not Antibacterial Activity, Is a Crucial Function of Hepcidins in Resistance against Pathogenic Bacteria in Nile Tilapia (*Oreochromis niloticus* Linn.)

**DOI:** 10.3390/biom10081132

**Published:** 2020-07-31

**Authors:** Pagaporn Phan-Aram, Gunanti Mahasri, Pattanapon Kayansamruaj, Piti Amparyup, Prapansak Srisapoome

**Affiliations:** 1Laboratory of Aquatic Animal Health Management, Department of Aquaculture, Faculty of Fisheries, Kasetsart University, 50 Paholayothin Rd, Ladyao, Chatuchak, Bangkok 10900, Thailand; pagaporn.ph@ku.th (P.P.-A.); pattanapon.k@ku.th (P.K.); 2Department of Fish Health Management and Aquaculture, Faculty of Fisheries and Marine, Universitas Airlangga, Campus C Mulyorejo, Surabaya 60115, Indonesia; mahasritot@gmail.com; 3Marine Biotechnology Research Team, Integrative Aquaculture Biotechnology Research Group, National Center for Genetic Engineering and Biotechnology (BIOTEC), National Science and Technology Development Agency (NSTDA), Pathum Thani 12120, Thailand; piti.amp@biotec.or.th

**Keywords:** nile tilapia, hepcidins, immune regulation, antimicrobial activity, *Streptococcus agalactiae*, *Flavobacterium columnare*

## Abstract

In this study, the functions of a recombinant propeptide (rPro*On*-Hep1) and the synthetic FITC-labelled mature peptides sMat*On*-Hep1 and sMat*On*-Hep2 were analyzed. Moreover, sMat*On*-Hep1 and sMat*On*-Hep2 were mildly detected in the lymphocytes of peripheral blood mononuclear cells (PBMCs) and strongly detected in head kidney macrophages. The in vitro binding and antibacterial activities of these peptides were slightly effective against several pathogenic bacteria. Immune regulation by sMat*On*-Hep1 was also analyzed, and only sMat*On*-Hep1 significantly enhanced the phagocytic index in vitro (*p* < 0.05). Interestingly, intraperitoneal injection of sMat*On*-Hep1 (10 or 100 µg) significantly elevated the phagocytic activity, phagocytic index, and lysozyme activity and clearly decreased the iron ion levels in the livers of the treated fish (*p* < 0.05). Additionally, sMat*On*-Hep1 enhanced the expression levels of *CC* and *CXC chemokines*, *transferrin* and both *On*-Hep genes in the liver, spleen and head kidney, for 1–96 h after injection, but did not properly protect the experimental fish from *S. agalactiae* infection after 7 days of treatment. However, the injection of *S. agalactiae* and *On*-Heps indicated that 100 μg of sMat*On*-Hep1 was very effective, while 100 μg of rPro*On*-Hep1 and sMat*On*-Hep2 demonstrated moderate protection. Therefore, *On*-Hep is a crucial iron-regulating molecule and a key immune regulator of disease resistance in Nile tilapia.

## 1. Introduction

Nile tilapia (*Oreochromis niloticus*) is the most important freshwater fish cultured in the world. The global production of farmed tilapia was approximately 6.5 million metric tons in 2019, and Thailand is among the most important producers, with an annual production of approximately 2 × 10^5^ tons [1]. In Thailand, Nile tilapia is practically produced by both cage culture and earthen pond systems, with a relatively high stocking density. This activity causes poor water quality and rapid changes in culture conditions. These effects directly increase stress and induce a number of infectious diseases caused by pathogenic bacteria, including *Aeromonas hydrophila*, *Flavobacterium columnare* and *Streptococcus agalactiae* [2]. Although antibiotic and chemical treatments are commonly used in the management of these fish diseases, the application of these agents is becoming increasingly limited, partially due to the potential health risks for consumers, but many fish pathogens have been associated with the development of antibiotic resistance.

An evaluation of fish immunology is required to overcome these problems, for the prevention and treatment of infectious diseases. Based on current information, vertebrates possess both innate and adaptive immunity. Innate immunity is the first line of defense against pathogens [3,4]. It is a nonspecific pathway, has limited memory, and is active against a variety of microbial agents. It crucially relies on both cellular and humoral responses. These responses produce variously effective proteins important for disease prevention mechanisms. Among these effective proteins, hepcidin has been discovered as a key molecule of the innate immune system. Structurally, a hepcidin molecule consists of the signal peptide and propeptide, which is subsequently cleaved to produce approximately 20–26 amino acid residues of mature peptide at the C-terminus. Previous studies have clearly demonstrated that it plays an important role in the innate immune system as an antimicrobial peptide. It is also considered to be an important hormone in iron homeostasis; iron is an essential element for many biological processes and required for the survival of all living organisms [5]. In teleost fish, the first hepcidin was identified in hybrid striped bass (*Morone chrysops* × *M. saxatilis*) in 2002 [6]. Since then, many different hepcidin molecules have been characterized, and their antimicrobial functions are well documented in fish species of great economic importance, including Atlantic salmon (Salmo salar) [7], zebrafish (Danio rerio) [8,9], channel catfish (*Ictalurus punctatus*) [10], tilapia (*Oreochromis mossambicus*) [11,12], marine medaka (*Oryzias melastigma*) [13], turbot (*Scophthalmus maximus*) [14,15], rainbow trout (*Oncorhynchus mykiss*) [5], Chinese rare minnow (*Gobiocypris rarus*) [16], spotted scat (*Scatophagus argus*) [17], roughskin sculpin (*Trachidermus fasciatus*) [18], large yellow croaker (*Larimichthys crocea*) [19] and goldfish (*Carassius auratus*) [20].

Little information is known about the immune function and regulation of hepcidins in Nile tilapia. Therefore, the purposes of this research were to overexpress and structurally analyze Nile tilapia hepcidin (*On*-Hep), as both a propeptide and a mature peptide, and to determine their antibacterial activities against pathogenic bacteria. Transcriptional level expression of *On*-Hep was quantitatively analyzed under pathogenic bacterial infection. The regulation of important immune-related gene transcription and iron levels in response to *On*-Hep application was also demonstrated. The antimicrobial activity of *On*-Hep1 was investigated in both in vitro and in vivo experiments. Importantly, the efficacy of *On*-Hep1 has been proven to act as an effective safeguard against harmful *S. agalactiae*. The information obtained from the current study is crucial for understanding the immune mechanisms and effective functional roles of *On*-Hep1, which is useful for generating prophylactic and therapeutic strategies to improve disease resistance against deadly pathogenic bacteria in the Nile tilapia aquaculture industry.

## 2. Materials and Methods

### 2.1. Characterization of the Full-Length cDNA Encoding On-Hep1

In this experiment, cDNA encoding uncharacterized On-Hep1 was retrieved from the cDNA library of Nile tilapia spleen deposited in the GenBank database (FF280957). The obtained cDNA sequence was reblasted against information available in the GenBank database (https://www.ncbi.nlm.nih.gov) using blastN and blastX programs. The 5′ UTR, ORF and 3′ UTR were analyzed by Genetyx 7.0. (Genetyx Co. Ltd., Tokyo, Japan) The signal peptide was predicted by the DAS transmembrane prediction server (http://www.sbc.su.se/~miklos/DAS/maindas.html). A homology analysis of *On*-Hep1 and other reported hepcidin gene sequences (Appendix A) by MatGAT version 2.01 (http://www.angelfire.com/nj2/arabidopsis/MatGAT.html) was conducted. Full-length *On*-Hep1 was aligned with other known tilapia hepcidin proteins using Genetyx 7.0.

### 2.2. Phylogenetic Analysis of On-Hep1 and Various Hepcidin Genes of Other Vertebrates

A phylogenetic tree of *On*-Hep1 and other hepcidin proteins found in various vertebrate species was constructed. All obtained sequences (Appendix A) were aligned using ClustalW (http://ebi.ac.uk/Tools/clustalw/index.html), and a neighbor-joining evolutionary tree was analyzed with 1000 bootstrapping values using MEGA version 6.0 (www.megasoftware.net).

### 2.3. Expression Analysis of On-Hep1 Transcripts in Various Tissues of Normal Nile Tilapia Using Quantitative Real-Time RT-PCR (qRT-PCR)

#### 2.3.1. Experimental Animals

Five hundred healthy Nile tilapia (84 ± 3.6 g) were obtained from Manit Genetics, Ltd., Phetchaburi Province, Thailand. The fish were acclimatized in a 500-L glass tank containing clean dechlorinated freshwater with an aeration system, for 7 days. During acclimatization, the fish were fed three times daily with commercial feed at 5% body weight. All experiments were conducted in accordance with the Ethical Principles and Guidelines for the Use of Animals recommended by the National Research Council of Thailand, for the care and use of animals for scientific purposes. The protocol was approved by the Animal Ethics Committee, Kasetsart University, Thailand, with the ethic ID of ACKU61-FIS-004.

#### 2.3.2. Total RNA Isolation and First-Strand cDNA Synthesis

Three fish were randomly collected and anesthetized with 80 mg/L clove oil (Hong Huat, Thailand), and whole blood was withdrawn from the caudal vein using a 2-mL heparinized syringe with a 21-G needle. PBLs were isolated with the method previously described by Chung and Secombes [21]. Subsequently, the fish were dissected, and the brain, gills, gonad, heart, head kidney, intestine, liver, muscle, skin, spleen, stomach and trunk kidney were collected. Total RNA from these 13 tissues was isolated using TRIzol reagent (Invitrogen, Waltham, MA, USA), according to the manufacturer’s instructions. The RNA pellet from all tissues was air-dried, and the total RNA was dissolved in sterile nuclease-free water. Total RNA samples were treated with DNase I (Fermentas, Pittsburg, PA, USA, USA) to remove contaminating genomic DNA. One microgram of total RNA from each tissue was separately used to synthesize first-strand cDNA with a RevertAid First Strand cDNA Synthesis kit (Fermentas, USA).

#### 2.3.3. qRT-PCR Analysis

A qRT-PCR analysis of *On*-Hep1 in each tissue was performed. One microliter of first-strand cDNA from each tissue was subjected to a Mx3005P real-time PCR system (Agilent Technologies, Inc., Santa Clara, CA, USA) equipped with analytical software version 4.0 (Agilent Technologies, Inc., Santa Clara, CA, USA) and Brilliant II SYBR Green qPCR Master Mix (Stratagene, USA), according to the manufacturer’s recommended protocol, using the specific primer pairs *On*-Hep1 F and *On*-Hep1 R (Appendix A). The *On*-Hep1 gene expression levels in each sample were normalized relative to the expression level of β-actin obtained using the primers *On*-β-actin F and *On*-β-actin R (Appendix A). The PCR conditions were 95 °C for 10 min, followed by 40 cycles of 95 °C for 30 s, 55 °C for 1 min and 72 °C for 1 min. A DNA melting curve analysis was used to verify the specificity of the primers. Triplicate reactions were performed for each tissue sample for the *On*-Hep1 and β-actin genes. A standard plasmid containing the On-Hep1 and β-actin genes was serially diluted in 10-fold increments to generate standard curves to assess PCR efficiency.

The threshold cycles (Ct) of the *On*-Hep1 and β-actin genes were measured, and the standard curve was used to determine their starting copy number. This method is based on equal PCR efficiencies for the target and internal control mRNA. The relative copy number of the target mRNA was calculated according to the 2-ΔΔCt method [22]. The threshold cycle value difference (ΔCt) between the *On*-Hep1 and β-actin mRNAs in each reaction was used to normalize the level of the total RNA. The relative expression of *On*-Hep1 in the brain was used as a calibrator.

### 2.4. Transcriptional Response Analysis of On-Hep1 in Liver, Spleen and Head Kidney under Flavobacterium columnare and Streptococcus Agalactiae Infection

#### 2.4.1. Bacterial Strains and Preparation

Two severely pathogenic bacteria, *Flavobacterium columnare* (AQFC001) and *S. agalactiae* (AQSA001), were obtained from the Laboratory of Aquatic Animal Health Management, Faculty of Fisheries. A single colony of *S. agalactiae* was cultured in 10 mL of trypticase soy broth (TSB) and incubated in a shaking incubator, at 30 °C for 24 h. The bacterial cells were centrifuged at 2500 rpm for 10 min, washed and resuspended in PBS (pH 7.4), adjusted to reach an absorbance of 0.669 at 600 nm, to obtain a concentration of 1 × 10^9^ CFU/mL, and further tenfold diluted with PBS to obtain a final concentration of 1 × 10^7^ CFU/mL. A single colony of *F. columnare* was inoculated with 10 mL of Shieh medium broth. Then, this bacterium was incubated and grown under the above conditions. The pellet was re-suspended in sterile water. The concentrations of *F. columnare* were adjusted to 1 × 10^7^ CFU/mL with optical densities of 0.27 at 525 nm. This bacterial suspension was used for other experiments below.

#### 2.4.2. Experimental Animals and Design

One hundred and fifty Nile tilapia in the above section were selected, placed in 5 250-L fiberglass tanks (30 fish each) and acclimatized as described above for 7 days. After that, all fish in tank 1 were intraperitoneally injected with 100 μL of PBS (pH 7.4), while fish in tanks 2-3 and 4-5 were injected with 100 μL of PBS (pH 7.4), containing 1 × 10^7^ and 1 × 10^9^ CFU/mL of *S. agalactiae* and *F. columnare*, respectively. These concentrations were selected based on a preliminary test that they could induce moderate and severe responses, days 5–7 after injection. At 0 h, 6 h, and 12 h and days 1, 2, 3 and 7, the liver, spleen and head kidney of 4 fish in each tank were collected for total RNA extraction, 1st-strand cDNA synthesis and qRT-PCR of *On*-Hep1 with the methods described above. The relative expression of On-Hep1 at h 0 was used as a calibrator for the transcription level analysis of each pathogenic induction.

### 2.5. Overexpression, Production and Purification of Recombinant On-Hep1 Propeptide (rProOn-Hep1)

#### 2.5.1. Construction of Recombinant rPro*On*-Hep1 DNA

Specific primers were designed to amplify the sequence encoding the Pro*On*-Hep1 peptide (ProOn-Hep1 F and Pro*On*-Hep1 R, Appendix A), which was subjected to PCR analysis using the following program: 95 °C for 5 min; 40 cycles of 95 °C for 30 s, 55 °C for 1 min, and 72 °C for 1 min; and an elongation step of 72 °C for 5 min. PCR products were run on a 1% agarose gel and further purified using the FavorPrep Gel/PCR Purification Mini Kit (Favorgen, Taiwan). The obtained PCR DNA fragment was ligated into the pGEM T-easy vector based on the provided protocol (Promega, Madison, WI, USA), which was then transformed into *Escherichia coli* (JM109) competent cells, using the heat-cold shock method. Blue-white colony screening was conducted on Luria–Bertani (LB) agar containing ampicillin (100 mg/mL), isopropyl-β-d-1-thiogalactopyranoside (IPTG) (1 mM) and X-gal (50 mg/mL). The plasmids were extracted from selected positive clones using the FavorPrep Plasmid Extraction Mini Kit (Favorgen, Ping-Tung, Taiwan) following the manufacturer’s protocol. Double restriction enzyme digestion with *Xho* I and *Nde* I was conducted following the recommendation of the company’s protocol (Thermo Scientific, Waltham, MA, USA), and the products were run on a 1% agarose gel. The desired digested DNA fragment was purified using the FavorPrep Gel/PCR Purification Mini Kit (Favorgen, Taiwan).

The purified DNA was ligated into a *Xho* I/*Nde* I-cut pET28b expression vector and transformed into JM109 competent cells as previously described. The transformed bacteria were selected and grown in LB broth (supplemented with 100 µg/mL kanamycin) at 37 °C. The positive clones were isolated, kept and extracted for plasmid DNA as described above. The obtained plasmid was sequenced with a similar method as described above, to check for the correct Pro*On*-Hep1 sequence. The rPro*On*-Hep1 plasmid was transformed into BL21 host cells on LB agar supplemented with 100 µg/mL kanamycin. The positive cells containing the rPro*On*-Hep1 plasmid were subcultured and kept in LB agar supplemented with 100 µg/mL kanamycin for further experiments.

#### 2.5.2. Overexpression of rPro*On*-Hep1 Using a Bacterial System

A single colony of BL21 cells containing the rPro*On*-Hep1 plasmid grown on LB agar plus 100 µg/mL kanamycin was inoculated into 3 mL of LB broth containing kanamycin (LBk) in a 15 mL test tube, and incubated at 30 °C overnight. Four hundred microliters of prepared bacterial preculture was transferred into 40 mL of LBk in a 50 mL conical PE tube incubated in a shaking incubator at 37 °C, for approximately 3 h. When the absorbance of the culture reached 0.6 at 600 nm, IPTG was added to a final concentration of up to 1 mM. The bacterial culture was further grown, and 1 mL of bacterial suspension was collected and put into 1.5 mL Eppendorf tubes at 1 h, 2 h, 3 h, 4 h and 5 h, and then centrifuged at 5000 rpm for 5 min. rPro*On*-Hep1 expression was determined using 12% SDS-PAGE, as described by Nakharuthai et al. (2017) [23].

#### 2.5.3. rPro*On*-Hep1 Purification

Based on the optimal induction times in the previous section, bacterial cells producing rPro*On*-Hep1 were grown and harvested in 500 mL Erlenmeyer flasks containing 250 mL of LBk under the previously described conditions. The target protein was purified by the Ni-NTA Purification System as described by the manufacturer with a modified protocol for an inclusion body protein (Invitrogen, Carlsbad, CA, USA). Finally, the recombinant protein was eluted by elution buffer (pH 8.0), and various protein fractions were collected into new Eppendorf tubes and characterized by SDS-PAGE techniques as described above. The obtained recombinant protein was dialyzed, and protein concentrations were determined by using the Bradford protein assay, compared with standard serial two-fold protein dilutions of 2 mg/mL albumin. All set concentrations were quantified for absorbance at 595 nm using an iMark™ Microplate Absorbance Reader (Bio-Rad, Drive Hercules, CA, USA). Total protein was also determined by SDS-PAGE using a Coomassie Protein Assay Kit (Thermo Scientific).

#### 2.5.4. Western Blot Analysis

Western blot analysis was used to confirm the existence and molecular weights of the obtained rPro*On*-Hep1 protein. The recombinant Pro*On*-Hep1 protein was separated on a 12% SDS-PAGE gel and further electrotransferred to a nitrocellulose membrane using a Mini Trans-Blot^®^ cell (Bio-Rad, Drive Hercules, CA, USA) at 0.11 A for 1 h and 10 min. The membrane was blocked with 5% skimmed milk at room temperature for 1 h, incubated with a 1:6000 dilution of a mouse anti-His tag antibody for 1 h, washed with 1X TBS + 0.1% Tween and incubated with a 1:5000 dilution of goat anti-mouse IgG-alkaline phosphatase (GAM-AP) for 1 h. The hybridized protein was detected using the Vectastain^®^ ABC kit (Vector Laboratories, Burlingame, CA, USA). The membrane was incubated with nitro blue tetrazolium (NBT)/ 5-bromo-4-chloro-3-indolyl-phosphate (BCIP) substrate and incubated with detection buffer (0.1 M Tris-HCl, 0.1 M NaCl and NBT 1 mL) for 30 min until color appeared.

### 2.6. Antimicrobial Activities of rProOn-Hep1, sMatOn-Hep1 and sMatOn-Hep2 (In Vitro)

Two commercially synthesized FITC-labeled mature peptides (sMat*On*-Hep1 and sMat*On*-Hep2) produced by China Peptides Co., Ltd. (China) and our own rPro*On*-Hep1 were used to evaluate MICs by liquid growth inhibition methods [24] for the four pathogenic bacteria *A. hydrophila* (AQAH009), *S. agalactiae* (AQSA001), *Vibrio parahaemolyticus* (AQVP001) and *V. vulnificus* (AQVV001). *A. hydrophila* and *S. agalactiae* were cultured in 10 mL of TSB medium and incubated in a shaking incubator at 30 °C for 24 h, while *V. parahaemolyticus* and *V. vulnificus* were grown in 10 mL of TSB supplemented with 1.5% NaCl, under conditions similar to those of the first two bacteria. The bacterial cells were centrifuged at 2500 rpm for 10 min, washed and resuspended in PBS (pH 7.4), adjusted to an absorbance of 0.669 at 600 nm to obtain a concentration of 1 × 10^9^ CFU/mL, and further tenfold diluted with PBS to obtain a final concentration of 1 × 10^8^ CFU/mL. Mueller–Hinton (MH) broth was used as an assay medium and supplemented with 1.5% NaCl for *V. parahaemolyticus* and *V. vulnificus* in a 96-well microtiter plate. The final concentrations of each peptide (128 to 0.25 μg/mL) in 200 μL of tested medium (in triplicate) were obtained by serial dilution, and bacterial concentrations of 1 × 10^5^ CFU were applied. Wells that contained only MH broth and bacteria were provided as a negative control. Plates were incubated at 30 °C, and the growth of bacteria in each well was followed by monitoring the absorbance at 600 nm at 0 h and then every 3 h until 24 h, using an iMark™ Microplate Absorbance Reader (Bio-Rad). After a 24 h incubation period, the lowest concentration that caused complete inhibition, as indicated by clear characteristics, was defined as the MIC value.

### 2.7. Effects of rProOn-Hep1, sMatOn-Hep1 and sMatOn-Hep2 on Phagocytosis of Peripheral Blood Mononuclear Cells (PBMCs) (In Vitro)

#### 2.7.1. Experimental Animal and Isolation of PBMCs

A healthy Nile tilapia (160.5 g) was used to isolate PBMCs using a heparinized syringe with a 23-G needle. One milliliter of whole blood was withdrawn from the caudal vein and put into a 15 mL PE tube containing 2 mL of RPMI medium (Caissonlabs, North Logan, UT, USA). Diluted blood was mixed gently, loaded into a new 15 mL PE tube containing 3 mL of LymphoprepTM (Serumwerk Bernburg AG, Oslo, Norway), and centrifuged in a swing rotor centrifuge at 400× *g* for 30 min. The A band, which consisted of monocytes, was harvested in 1 mL, that was mixed with 2 mL of PBS (pH 7.4) and then centrifuged at 200× *g* for 10 min, twice. The obtained cells were numerated using a hemocytometer and further diluted to reach a final concentration of 5 × 10^6^ cells/mL.

#### 2.7.2. Phagocytic Activity Analysis

Two hundred microliters of the above-prepared leukocytes containing phagocytes were loaded onto eight 22 × 22 mm^2^ cover glasses and allowed to adhere on the glass surface for 2 h. Monolayer cells were washed 3 times with PBS (pH 7.4) to remove unattached cells. Previously, 200 µL of PBS (pH 7.4) containing 1 × 10^7^ latex beads (Sigma-Aldrich, St. Louis, MO, USA) supplemented with 100 µg of each rPro*On*-Hep1, sMat*On*-Hep1 and sMat*On*-Hep2 was prepared for 1 h. Two hundred microliters of three latex bead types, each incubated with a different peptide, were added onto the cover glass with each duplicate. After 1.5 h, the unattached cells and excess beads were washed 3 times with PBS. The other two cover glasses with only 200 µL of PBS containing latex beads were provided as a control group. All attached cells in each treatment were stained with a Dip-Quick Staining Kit according to the manufacturer’s recommendations (Vetanymall, Nonthaburi, Thailand). PA was observed under a microscope by counting 100 cells in each cover glass, and percentages of PA and PI were calculated as described in a previously modified method [25], following these 2 Formulas (1) and (2):Percent PA = (number of cells with engulfed latex beads) × 100/(number of phagocytes)(1)
PI = number of engulfed latex beads/number of phagocytic cells(2)

### 2.8. Pathogenic Bacterial Binding Activity of sMatOn-Hep1 and sMatOn-Hep2 (In Vitro)

#### 2.8.1. Preparation of Pathogenic Bacteria

In this experiment, the three pathogenic bacteria *A. hydrophila*, *Flavobacterium columnare*, *S. agalactiae* and the nonpathogenic species *Bacillus pumilus* were used. All bacteria were cultured and prepared as described above to obtain a final concentration of 1 × 10^8^ CFU/mL in a bacterial suspension. One hundred microliters of each bacterial suspension were separately mixed and incubated with 100 µL of PBS, containing 100 µg of the FITC-labeled peptides sMat*On*-Hep1 and sMat*On*-Hep2 for 2 h at ambient room temperature (25 °C).

#### 2.8.2. Cell Binding Analysis

The cells were washed with PBS and centrifuged at 400× *g* for 10 min three times. After the cells were washed, 10 μL of the cell-peptide combinations were dropped onto a glass slide, and the binding activities of each peptide were observed under a confocal microscope (Nikon Eclipse Ti, Nikon C2 confocal microscope, Melville, NY, USA) using 100× magnification.

### 2.9. Localization of sMatOn-Hep1 and sMatOn-Hep2 on Leukocytes from PBMCs and Head Kidney of Nile Tilapia (In Vitro)

#### 2.9.1. Preparation of Blood Leukocytes

A healthy Nile tilapia (160.5 g) was isolated for PBMCs based on the above description, and leukocytes from head kidney were isolated using the modified method described by Ortuno et al. (2001) [26]. Cells were adjusted with PBS (pH 7.4), to a final concentration of 1 × 10^8^ cells/mL.

#### 2.9.2. Localization Assay

One hundred microliters of diluted cell suspension were mixed with 100 µL of each FITC-labeled peptide, i.e., sMat*On*-Hep1 and sMat*On*-Hep2 (100 µg for each), for 2 h at ambient room temperature (25 °C). The cells were washed and centrifuged, and cell binding activity was observed with the same practices as described in the section above.

### 2.10. Effects of rProOn-Hep1 on Resistance to Streptococcus agalactiae in Nile Tilapia (In Vivo)

#### 2.10.1. Experimental Animals

One hundred and forty-eight healthy Nile tilapia (70.6 ± 4.7 g) were acclimatized in a 1000-L fiberglass tank containing dechlorinated freshwater, with full aeration conditions, for 7 days. Then, each set of 48 fish was placed into separate 500-L fiberglass tanks containing 400 L of freshwater. During this time, fish were fed twice a day with commercial feed at 5% body weight, and 20% of the water was changed daily.

#### 2.10.2. Preparation of *Streptococcus agalactiae* and rProOn-Hep1

Virulent *S. agalactiae* strains were cultured and prepared as described above to obtain a bacterial suspension at a final concentration of 1 × 10^8^ CFU/mL using PBS (pH 7.4). PBS was used to dilute rProOn-Hep1 solution from a previous section to the 3 concentrations 10, 50 and 100 µg rProOn-Hep1/100 mL.

#### 2.10.3. Bacterial Challenge

The experiment was conducted and run with 3 replicates under a cohabitation culture system. Each of four groups of twelve fish in each tank from a previous section were differently tagged with one of four colored threads at the first spine of the dorsal fin. Each experimental group was intraperitoneally injected with 100 µL of bacterial suspension. One hour later, a second injection of 100 µL, containing 1 of 3 different concentrations of rPro*On*-Hep1, was performed as described in a previous section. The fourth group of 12 fish in each tank were similarly injected with 100 µL PBS to serve as controls. Fish continued rearing, and mortality was recorded daily for 10 days. Moribund fish found in each tank were collected and diagnosed for bacterial infection using the liver and spleen by the loop isolation method in TSA.

### 2.11. Effects of sMatOn-Hep1 and sMatOn-Hep2 on Resistance against Streptococcus agalactiae in Nile Tilapia (In Vivo)

#### 2.11.1. Experimental Animals

One hundred healthy Nile tilapia (100.5 ± 9.3 g) were acclimatized in the same environment in a 1000-L fiberglass tank as described above, with slight modification. Then, 50 fish each were brought into a 500-L fiberglass tank containing 400 L of freshwater.

#### 2.11.2. Preparation of *S. agalactiae* Solution and sMatOn-Hep1 and sMat*On*-Hep2 Peptide

The virulent *S. agalactiae* strains were cultured and prepared as described above. Two commercially synthesized FITC-labeled sMat*On*-Hep1 and sMat*On*-Hep2 peptides from the above section were aliquoted at the 2 concentrations (diluted with PBS) 10 µg and 100 µg peptides/100 μL. The bacterial suspension was diluted with PBS to provide 1 × 10^8^ CFU of *S. agalactiae*.

#### 2.11.3. Bacterial Challenge

This experiment was also conducted in cohabitation culture, and 10 fish in each tank were tagged by 5 differently colored threads. Ten fish from each group were intraperitoneally injected with 100 µL of a bacterial suspension. One hour later, 2 different concentrations of sMat*On*-Hep1 and sMatOn-Hep2 prepared as described above were injected under the same conditions. The other 10 fish in each tank served as controls and were injected with 100 µL of a bacterial suspension. Fish were observed and recorded for mortality until day 10, with a similar procedure, as described above.

### 2.12. Effects of the sMatOn-Hep1 Peptide on Immune Responses and Iron, Zinc and Copper Concentrations in Nile Tilapia

#### 2.12.1. Preparation of sMatOn-Hep1 Peptide

Commercially synthesized FITC-labeled sMat*On*-Hep1 peptide from the above section was aliquoted and mixed with PBS (pH 7.4) to provide the 2 concentrations 10 and 100 µg sMat*On*-Hep1/100 μL.

#### 2.12.2. Experimental Animals and Experimental Design

Ninety healthy Nile tilapia (130.3 ± 6.5 g) were acclimatized in a 1000-L fiberglass tank containing dechlorinated freshwater with full aeration conditions for 7 days. Then, each group of 30 fish was placed into separate 500-L fiberglass tanks containing 400 L of freshwater to set up 3 experimental groups. Each group of ten fish in each tank was tagged with 1 of 3 different colors of thread. After acclimatization, 2 groups of ten fish each in each tank were intraperitoneally injected with 100 µL of sMatOn-Hep1 solution prepared in the previous section. The other 10 fish in each tank served as the control and were injected with 100 µL of PBS (pH 7.4). At the beginning of the experiment and prior to peptide injection, whole blood and liver of each fish were collected from each tank, to investigate innate immune parameters and heavy metal concentrations, including iron, zinc and copper. PBMCs of blood samples were isolated as described above. After injection, the same investigations of a fish in each replicate of each treatment were conducted at h 0, 1, 6, 12, 24, 48 and 96.

#### 2.12.3. Innate Immune Parameter Analysis

##### Phagocytic Activity

PBMCs were isolated from 3 fish in each treatment at 0, 1, 6, 12, 24, 48 and 96 h, as described in a previous section. The densities of the latex beads (Sigma-Aldrich, USA) and phagocytic cells were adjusted to 1 × 10^8^ beads/mL and 5 × 10^6^ cells/mL, respectively, with RPMI medium. The phagocytic assay was conducted with the same protocol as described in a previous section.

##### Lysozyme Activity

In this experiment, another fish from each treatment at different time points similar to those in the above section was withdrawn, and its whole blood was collected with the same method as described above, transferred to a new Eppendorf tube and rested at ambient temperature for 2 h. Serum was separated from the blood corpuscles by centrifugation at 3500 rpm for 15 min. Approximately 500 µL was transferred and kept in a new tube. A suspension of *Micrococcus lysodeikticus* (200 µg/mL) in PBS (pH 6.2) was prepared next. Ten microliters of fish serum were added to 3 wells of a 96-well flat-bottom microtiter plate containing 250 µL of prepared *M. lysodeikticus*. The other 3 wells received 10 µL of PBS (pH 6.2) to serve as a control. The absorbance of the reaction wells was measured at 540 nm after 1 and 5 min using an iMark™ Microplate Absorbance Reader (Bio-Rad). Lysozyme activity was calculated with the method described by Sarder et al. [25] The serum used in this study was also utilized to analyze complement activity and metal element content below.

##### Alternative Complement Activity (ACH_50_)

One hundred microliters of PBS buffer (pH 7.4) was placed into 10 wells of a 96-well U-bottom microplate. Serial two-fold dilutions were conducted by adding 100 µL of each fish serum from the first to the last well. Unlysed and completely lysed controls were set by using 100 µL of PBS buffer (pH 7.4) and distilled water, respectively. Then, 100 µL of sheep red blood cells (CLINAG Co., Ltd., Bangkok, Thailand) in PBS buffer (pH 7.4) at a concentration of 2 × 10^8^ cells/mL were loaded into every well. The reaction mixtures were incubated for 1 h at 25 °C and centrifuged at 1500× *g* for 5 min at 25 °C. Finally, 200 μL of supernatant in each well was transferred to a new 96-well flat-bottom microplate, and the absorbance was measured at 540 nm using an iMark™ Microplate Absorbance Reader (Bio-Rad). Complement activity in serum samples in terms of ACH_50_ was analyzed based on the modified method of Ortuno et al. [26].

##### Concentrations of Iron, Zinc and Copper in the Liver, Blood Corpuscles and Serum

Fish livers and blood corpuscles (1 g) were cut into small pieces and predigested in a Teflon beaker with 10.0 mL of HNO_3_ (68%, acid solution). The digested sample in acid solution was transferred to a Teflon PFA pressure decomposition vessel (CEM Corporation, Matthews, NC, USA) and heated in a microwave digester (MARS 6, CEM Corporation) at 200 °C for 15 min. In the case of blood serum, 0.5 mL of serum was used and put in a Teflon beaker, containing 1.25 and 0.5 mL of concentrated HNO_3_ and hydrogen peroxide, respectively. Mixed components were heated under the same conditions as in the abovementioned methods. The resulting transparent liquid of the digested liver, blood corpuscles and serum was moved to a Teflon beaker. Fe, Zn and Cu elements were analyzed by atomic absorption spectroscopy (AAS, Agilent 240FS AA, Agilent Technologies, Inc., Santa Clara, CA, USA). Standard solutions of each element were prepared by following the methods described by PanReac AppliChem ITW (ITW Reagents Division, Barcelona, Spain), and used to obtain calibration curves. Each sample solution was analyzed three times with AAS to determine the 3 target elements.

### 2.13. Effects of sMatOn-Hep1 on the Regulation of Immune-Related Genes Using qRT-PCR and Bacterial Resistance

In this experiment, the same fish used for immune analysis in the above section were dissected to collect the liver, spleen and head kidney. Total RNA of these tissues was isolated, and first-strand cDNA was synthesized as described in a previous section. The obtained first-strand cDNA was separately subjected to qRT-PCR analysis under the same conditions as described above, using specific primers for 7 immune-related genes of Nile tilapia: *CC chemokine-1*, *CC chemokine-2*, *interleukin-8* (*On*-CXC1 and *On*-CXC2), *interleukin-1β*, *hepcidin1* (*On*-Hep1), *hepcidin2* (*On*-Hep2) and *transferrin* (Appendix A).

#### 2.13.1. Challenge Analysis

##### Preparation of Virulent *S. agalactiae*

Pathogenic *S. agalactiae* (AQSA001) was prepared with the same system as described in the previous section to reach a final concentration of 1 × 10^8^ CFU/mL.

##### Challenge Procedure

After sMat*On*-Hep1 application for 7 days, the remaining fish in each treatment from different tanks in the above section were intraperitoneally injected with 100 µL of bacterial suspension prepared in the previous part. The fish in each tank received good care for 10 days. The mortality assessments and disease diagnoses of the moribund fish were conducted daily, as described above, until day 10 of injection.

### 2.14. Statistical Analysis

The relative expression ratios of *On*-Hep1 in qRT-PCR analyses, immune parameters and concentrations of 3 heavy metals were statistically analyzed using analysis of variance (ANOVA), and the means of these parameters were compared by Duncan’s new multiple range test (DMRT) using SPSS software version 20.0 (IBM). A survival analysis in the challenge experiments in each group was performed using the Kaplan–Meier method with the same software. *p* < 0.05 was considered statistically significant.

## 3. Results

### 3.1. Characterization of the Full-Length cDNA of On-Hep1

The full-length cDNA encoding the hepcidin gene of Nile tilapia was successfully characterized and named *On*-Hep1. It comprised 672 bp, including a 114 bp 5′ untranslated region (UTR), a 273 bp open reading frame (ORF) and a 285 bp 3′ UTR with a polyadenylation site (AATAAA). The predicted amino acid sequence of the propeptide (pro*On*-Hep1) contained 90 amino acid residues. A signal prediction program showed that pro*On*-Hep1 had a signal peptide and a propeptide, ranging from amino acids 1–20 and 21–90, respectively (Figure 1a–c). The mature peptide (m*On*-Hep1), containing 26 amino acids, was found at the C terminus, with eight cysteine residues that are highly conserved among the other Cichlid hepcidin proteins (Figure 1b).

Homology analysis revealed that *On*-Hep1 shares sequence similarity and identity with hepcidin genes in other fish species and higher vertebrates, that range from 43.2–97.4% and from 43.1–97.4%, respectively. The highest similarity and identity were shared with Mozambique tilapia (*Oreochromis mossambicus*) hepcidin TH 2-3. In comparison, the *On*-Hep2 similarity and identity were 68.1 and 67.4%, respectively (Appendix A). Additionally, the possible iron regulatory sequence “QSHLSL” was evidenced on the *On*-Hep1 molecules, similar to *Om*-TH2-3 of Mozambique tilapia.

### 3.2. Phylogenetic Analysis of Hepcidin Genes in Vertebrates

Thirty-four hepcidin proteins from various vertebrates and the *On*-Hep protein were used to construct a neighbor-joining phylogenetic tree. Three major clusters of hepcidin genes were clearly identified. Clades 1 and 2 belonged to fish hepcidins, while clade 3 was totally occupied by hepcidins of higher vertebrates. Clade 1 included only teleost hepcidin and contained 3 different minor groups of teleost hepcidins 1 and 2 and grouper hepcidins 1 and 2. *On*-Hep1 and *On*-Hep2 were grouped in the second and first minor groups of clade 1, respectively. *On*-Hep1 showed strong relationships with type I hepcidins in Mozambique tilapia, Pacific mutton hamlets, Asian sea bass, largemouth black bass and smallmouth bass. Grouper hepcidins 1 and 2 were solely located in different minor groups of clade 1. Clades 2 and 3 clearly included an additional hepcidin group from teleosts and higher vertebrate hepcidin, respectively (Appendix A).

### 3.3. Distribution of On-Hep1 Transcripts in Various Tissues of Healthy Nile Tilapia

qRT-PCR analysis was used to analyze the tissue distribution of Nile tilapia *On*-Hep1 mRNA expression in 13 different tissues of 3 normal fish. The analysis showed that *On*-Hep1 was expressed at the highest level in the liver, at a level 795 ± 35.1-fold greater than the level in the brain, while moderate expression was observed in the spleen, head kidney and heart at 174.33 ± 21.7-, 58.44 ± 19.4- and 56.64 ± 13.1-fold the brain level, respectively. Very low levels were observed in the brain, gills, gonad, intestine, muscle, PBLs, skin, stomach and trunk kidney (Figure 2).

### 3.4. Analysis of Transcriptional Response of On-Hep1 in Liver, Spleen and Head Kidney under S. agalactiae and F. columnare Infection

In the liver, significant relative expression levels of *On*-Hep1 were observed at 6 h and 12 h after injection with *S. agalactiae* at 1 × 10^7^ and 1 × 10^9^ CFU/mL, which resulted in 123.2 ± 20.4- and 341.9 ± 18.2-fold changes, respectively (*p* < 0.05) (Figure 3A). At days 1–7, the expression levels of the target gene were lower than these levels, but still significantly differed from those of the control in some periods. However, the highest concentration of *F. columnare* significantly induced the expression of *On*-Hep1 at only 6 h and day 7, with expression levels 82.4 ± 9.3- and 74.1 ± 9.1-fold that of the control, respectively (Figure 3B).

In the spleen, with the same trend as that in the liver, the highest dose of *S. agalactiae* was significantly effective, increasing *On*-Hep1 expression at 6 h and 12 h after injection, by 45.9 ± 6.1- and 438.1 ± 2.2-fold, respectively (Figure 3C) (*p* < 0.05). In an *F. columnare* injection experiment, the highest concentration elevated upregulation of the target gene at 6 h by only 21.5 ± 1.8-fold (Figure 3D). Similar to the previously tested tissues, the highest concentration of *S. agalactiae* strongly induced *On*-Hep1 expression in the head kidney at 6 h and 12 h and day 1 after injection, when its level increased 22.8 ± 1.0, 545.1 ± 13.6- and 156.9 ± 4.9-fold, respectively (Figure 3E), and the highest concentration of *F. columnare* typically induced *On*-Hep1 expression at 6 h, increasing its expression by 193.4 ± 7.6-fold (*p* < 0.05). No obvious upregulation of the target gene was observed after this period (Figure 3F).

### 3.5. Characterization of rProOn-Hep1

rPro*On*-Hep1 was successfully overexpressed in *E. coli* BL21 cells. rPro*On*-Hep1 had a molecular weight of approximately 10 kDa, and most of it formed inclusion bodies (Appendix A). After it was denatured with 8 M urea, the soluble form of the target protein was successfully purified (Appendix A). Western blot analysis confirmed the presence of the rPro*On*-Hep1 protein, which specifically bound to an anti-His-tag monoclonal antibody (Figure 4A,B).

### 3.6. Functional Analyses of rProOn-Hep1, sMatOn-Hep1 and sMatOn-Hep2 in Controlling Pathogenic Bacteria

In this experiment, the efficacies of rPro*On*-Hep1, sMat*On*-Hep1 and sMat*On*-Hep2 in controlling four pathogenic bacteria were tested to determine the minimal inhibitory concentration (MIC), but no MIC values were clearly observed based on the criteria. However, based on their detected absorbance, some concentrations of rPro*On*-Hep1 showed mild antibacterial activity against *A. hydrophila*, *S. agalactiae* and *V. vulnificus* (Figure 5A,B,D, respectively). Furthermore, sMat*On*-Hep1 and sMat*On*-Hep2 did not clearly inhibit the growth of the 4 pathogenic bacteria (Figure 5E–L).

### 3.7. Effects of rProOn-Hep1, sMatOn-Hep1 and sMatOn-Hep2 on Phagocytic Activity (In Vitro)

The effects of rPro*On*-Hep1, sMat*On*-Hep1 and sMat*On*-Hep2 on the phagocytic activity (PA) of Nile tilapia phagocytes were examined. It was shown that 100 µg/mL of these proteins could not enhance the PA of Nile tilapia in vitro compared with that of the control (Figure 6A). However, sMat*On*-Hep1 enhanced the phagocytic index to a value significantly higher (2.2 ± 0.11) than those of the control and other peptides, which resulted in indices of 1.84 ± 0.03, 2.08 ± 0.04 and 2.01 ± 0.12, respectively (*p* < 0.05) (Figure 6B).

### 3.8. sMatOn-Hep1 and sMatOn-Hep2 Binding of Bacterial Cells

The bacterial cell binding of sMat*On*-Hep1 and sMat*On*-Hep2 were investigated (Figure 7A–H). These two peptides exhibited binding activity at different levels. Both peptides demonstrated no reaction with *B. pumilus* (Figure 7C,D). The sMat*On*-Hep1 clearly formed many aggregates by binding *F. columnare* (Figure 7E), while such results were not observed in the sMat*On*-Hep2 experiment (Figure 7F). On the other hand, mild binding was observed between sMat*On*-Hep2 and *A. hydrophila* (Figure 7B), while very little binding occurred between this bacterium and sMat*On*-Hep1 (Figure 7A). Finally, *S. agalactiae* cells adhered to form many clumps with sMat*On*-Hep1 (Figure 7G), while sMat*On*-Hep2 reacted with only a few single, round cells (Figure 7H).

### 3.9. Localization of sMatOn-Hep1 and sMatOn-Hep2 in PBMCs and Leukocytes from Head Kidney

In this experiment, FITC-labeled sMat*On*-Hep1 and sMat*On*-Hep2 peptides showed rare binding activity on the surface membrane of Nile tilapia peripheral blood mononuclear cells (PBMCs) (Figure 8A,B, respectively), and signals from very few cells were observed. Additionally, many clumps of proteins and cells appeared in the sMat*On*-Hep1 reaction. Moreover, sMat*On*-Hep1 was clearly observed in the cytoplasm of several head kidney leukocytes (Figure 8C), whereas sMat*On*-Hep2 showed a weaker signal in fewer cells (Figure 8D).

### 3.10. Efficacy of rProOn-Hep1 on S. agalactiae Resistance in Nile Tilapia

In this experiment, the efficacy of rPro*On*-Hep1 in protecting Nile tilapia from *S. agalactiae* in cohabitating conditions was evaluated. The results revealed that, at days 5–10, fish injected with 100 µg of rPro*On*-Hep1 showed significantly lower mortality (58.3 ± 8.3% less) than the control and other treated groups, which exhibited mortality rates of 91.7 ± 8.3%, 77.8 ± 9.6% and 80.6 ± 9.6% (*p* < 0.05) (Figure 9A).

A survival analysis of the challenge test in each group was performed using the Kaplan–Meier method.

### 3.11. Efficacy of sMatOn-Hep1 and sMatOn-Hep2 on S. agalactiae Resistance in Nile Tilapia

The efficacy of sMat*On*-Hep1 and sMat*On*-Hep2 in controlling *S. agalactiae* was tested in cohabitating conditions. The results in Figure 9B show that, at days 4–10, fish injected with 100 µg of sMat*On*-Hep1 demonstrated significantly lower mortality (10.0 ± 0.0%) than the control and other fish (*p* < 0.05). During these periods, fish injected with sMat*On*-Hep1 (10 µg) or sMat*On*-Hep2 (100 µg) also showed significantly lower mortality (30.0 ± 0.0% and 25.00 ± 7.07%) than fish injected with sMat*On*-Hep2 (10 µg) or the control, which had mortality rates of 50.0 ± 0.0% and 75.0 ± 7.1%, respectively (*p* < 0.05). The mortality of the control group was significantly higher than those of the other groups (*p* < 0.05).

### 3.12. Effects of the sMatOn-Hep1 Peptide on Immune Responses and Heavy Metal Concentrations

#### 3.12.1. Phagocytic Activity

To examine the effect of sMat*On*-Hep1 on the PA of Nile tilapia phagocytes, an in vitro experiment was designed. At 48 h, just 100 µg of this peptide could significantly enhance (78.0 ± 2.0%) the PA of Nile tilapia compared with those of the control and 10 µg sMat*On*-Hep1-injected group (71.0 ± 4.0% and 72.0 ± 2.0%, respectively) (*p* < 0.05) (Figure 10A). However, both 10 and 100 µg of sMat*On*-Hep1 enhanced the phagocytic index (PI), which was significantly higher for these fish than for the control at 12 h and 24 h (*p* < 0.05), with values of 1.79 ± 0.04, 1.77 ± 0.14, 1.43 ± 0.03, 1.94 ± 0.31, 2.22 ± 0.05 and 1.53 ± 0.07 (Figure 10B). Furthermore, at 48 h, only the group of fish induced with 100 µg of sMat*On*-Hep1 demonstrated a PI (2.79 ± 0.40) significantly higher than that in the control and other treated groups by 1.43 ± 0.10 and 1.93 ± 0.32, respectively (*p* < 0.05).

#### 3.12.2. Lysozyme Activity

It was shown that, at 24 h and 48 h, 10 µg and 100 µg of sMat*On*-Hep1 significantly induced serum lysozyme activity in Nile tilapia (*p* < 0.05). The highest induction was observed at h 48, and fish stimulated with 10 and 100 µg of sMat*On*-Hep1 expressed lysozyme activities of 310.00 ± 10.00 and 240.00 ± 17.32 units/mL, respectively, which were significantly higher than that of the control (136.67 ± 32.15 units/mL) (*p* < 0.05) (Figure 10C).

#### 3.12.3. Alternative Complement Activity (ACH_50_)

The ACH_50_ in fish serum was investigated. However, no significant difference between fish groups was observed in any period of time (*p* > 0.05), with the concentration ranging from 91.37 ± 38.35-215.08 ± 60.67 units/mL (Figure 10D).

#### 3.12.4. Concentrations of Iron, Zinc and Copper in the Liver, Blood Corpuscles and Serum

To study the effects of sMat*On*-Hep1 on the regulation of iron (Fe), zinc (Zn) and copper (Cu), the concentrations of these metal elements were quantified in the liver, blood corpuscle and serum of stimulated fish.

For Fe, application of the target peptide at 100 µg clearly reduced Fe concentration in liver and blood corpuscles at 6 h after injection by 64.1 ± 25.0, 79.5 ± 4.2, 111.9 ± 26.8 and 121.1 ± 25.1, 158.1 ± 7.7, 190.2 ± 19.3 µg/g, respectively (Figure 11A,B). However, at 24 h, the two concentrations of sMat*On*-Hep1 used significantly increased Fe concentrations (68.5 ± 3.0 and 73.2 ± 10.5 µg/g) in the liver, higher than those of the control (39.7 ± 2.7 µg/g). Moreover, at 48 h and 96 h, only 100 µg of peptides resulted in significantly higher Fe (79.3 ± 15.5 and 81.7 ± 14.4 µg/g) than the control and other treated group (*p* < 0.05), with consistent levels of 54.9 ± 6.6 and 45.1 ± 6.6, and 59.9 ± 9.1 and 38.4 ± 10.0 µg/g at the two respective times (Figure 11A). No significant differences in Fe levels were observed in serum, even though they increased in every group by approximately two times at 48 h and 96 h (*p* > 0.05) (Figure 11C). For Zn concentrations, no significant differences in Zn levels were observed in the blood corpuscles and serum of fish during the experimental period (Figure 11D,E). However, 10 and 100 µg of sMat*On*-Hep1 (10.2 ± 0.2 and 11.0 ± 1.0 µg/g) could significantly maintain Zn levels in fish liver at 96 h after injection, compared with the control (8.5 ± 0.3 µg/g) (*p* < 0.05) (Figure 11F). Regarding the Cu concentrations, the 10-µg concentration of sMat*On*-Hep1 was found to rapidly decrease the Cu levels at 6 h in the liver by 35.5 ± 3.6 µg/g compared to the control and 100 µg of sMat*On*-Hep1 treatment (52.2 ± 3.5 and 59.3 ± 11.9 µg/g) (Figure 11G). sMat*On*-Hep1 at 100 µg significantly increased the Cu levels by 1.6 ± 0.4 µg/g compared to the control and 10 µg of sMat*On*-Hep1 treatment (0.4 ± 0.0 and 0.2 ± 0.0 µg/g) (*p* < 0.05) (Figure 11H). The concentration of Cu in serum was lower than the detectable level in every time interval (data not shown).

### 3.13. Regulation Analysis of Immune-Related Genes in Nile Tilapia, Stimulated by sMatOn-Hep1 Protein

In this experiment, the effects of sMat*On*-Hep1 on the expression of eight immune-related genes, including *CC chemokine1* (*On*-CC1), *CC chemokine2* (*On*-CC2), *CXC chemokine1* (*On*-CXC1), *CXC chemikine2* (*On*-CXC2), *interleukin 1b* (*On*-IL 1b), *hepcidin1* (*On*-Hep1, itself), *hepcidin2* (*On*-Hep2) and *transferrin* (*On*-Trans), in three different organs were investigated by qRT-PCR techniques.

In the head kidney (Figure 12A–H), very quick responses at the first hour were observed in all immune-related genes, except transferrin. It was found that 100 µg of sMat*On*-Hep1 significantly enhanced the upregulation of these genes (*p* < 0.05) compared to their levels in the control fish by 75.4 ± 20.4-, 13.9 ± 4.7-, 2.6 ± 0.8-, 27.7 ± 13.4-, 2.7 ± 1.7-, 5.4 ± 3.3- and 22.2 ± 6.4-fold. At 6 h, this concentration of sMat*On*-Hep1 further significantly upregulated *On*-CC1, *On*-CC2, *On*-Hep1 and *On*-Hep2, by 33.3 ± 6.5-, 11.8 ± 6.7-, 13.6 ± 8.1- and 11.4 ± 2.6-fold, respectively. Additionally, both 10 and 100 µg of sMatOn-Hep1 significantly enhanced the upregulation of *On*-CC1 at 12 h, *On*-CC2 at 48 h and *On*-CXC1 at 24 h, compared to their levels in the control group (27.8 ± 7.0 and 31.9 ± 19.3, 20.6 ± 9.3 and 19.7 ± 1.7, and 6.0 ± 2.7 and 10.1 ± 2.7, respectively). However, the downregulated expression of these genes caused by the two concentrations of sMat*On*-Hep1 was observed at 96 h after injection.

In the liver (Figure 13A–H), clearly, significant *On*-CC1 and *On*-Hep1 upregulation caused by injection with only 100 µg of sMat*On*-Hep1 was demonstrated at 48 h, increasing their levels by 7.1 ± 4.3- and 400.3 ± 184.0-fold (*p* < 0.05), respectively. Ten micrograms of sMat*On*-Hep1 strongly enhanced the expression of *On*-CXC1 and *On*-Hep1 at only 12 h, by 6799.3 ± 559.2- and 100.8 ± 31.1-fold, respectively. This concentration also significantly upregulated the expression of *On*-CC2, *On*-CXC2 and *On*-Trans at 12 h and 96 h after injection, resulting in values of 26.4 ± 8.3 and 6.2 ± 0.6, 9.2 ± 3.6 and 16.3 ± 6.8, and 45.3 ± 9.5 and 81.2 ± 26.7, respectively. Both injected concentrations (10 µg and 100 µg) of sMat*On*-Hep1 effectively increased the expression fold of *On*-CC2, *On*-CXC2, *On*-IL1b, *On*-Hep2 and *On*-Trans expression (*p* < 0.05) at 48 h (5.2 ± 2.0, 9.4 ± 2.0 and 14.1 ± 5.3; 0.9 ± 0.4, 15.9 ± 4.7 and 81.8 ± 18.5; 0.4 ± 0.1, 2.7 ± 0.7 and 5.3 ± 1.8; 23.7 ± 12.0, 24.9 ± 5.9 and 312.9 ± 102.4; and 11.2 ± 2.5, 66.7 ± 35.8 and 150.0 ± 88.4, respectively).

In the spleen (Figure 14A–H), fish injected with 10 µg and 100 µg of sMat*On*-Hep1 were found to have significantly downregulated *On*-CC2, *On*-CXC1, *On*-IL1b, *On*-Hep1 and *On*-Trans 1–96 h after injection (*p* < 0.05). In addition, 100 µg of sMat*On*-Hep1 significantly increased the expression level of *On*-CC1 at h 6 by 16.8 ± 3.5-fold, and 10 and 100 µg upregulated *On*-CC1 expression at h 24 by 6.5 ± 1.4- and 6.5 ± 2.9-fold, respectively, while the control was expressed at approximately 1.2 ± 0.8. Interestingly, it was clear that the application of 10 and 100 µg of sMat*On*-Hep1 strongly upregulated the expression of *On*-CXC2 compared to that of control fish, at 12 h, 24 h, 48 h and 96 h after injection by 0.18 ± 0.3, 295.2 ± 40.3 and 487.5 ± 182.8; 22.9 ± 9.3, 86.8 ± 41.2 and 631.3 ± 84.0; 35.6 ± 21.6, 320.9 ± 226.4 and 833.9 ± 217.7; and 11.46 ± 6.4, 356.4 ± 134.9 and 456.6 ± 174.6. In addition, 100 µg of sMat*On*-Hep1 clearly significantly upregulated *On*-Hep2 expression, compared with that of other tested groups (*p* < 0.05) at 1 h, 24 h and 96 h, resulting in values of 28.7 ± 15.3, 31.3 ± 12.3 and 94.2 ± 7.8, respectively.

### 3.14. Challenge Analysis

After fish were injected with sMat*On*-Hep1 for 7 days; at days 1 and 2 after *S. agalactiae* injection, only the fish group that were given 100 µg of peptide showed high resistance to infection with no mortality, while the other groups exhibited rapid increases in mortality in these periods (*p* < 0.05) (Figure 15). However, at days 3–5, the mortality rate of each group increased very quickly, and at day 5, significant differences were observed between only the control and 100 µg/100 µL sMat*On*-Hep1-injected groups (*p* < 0.05). However, at days 6–10, the mortality rate of each group reached the highest points, at 86.7 ± 11.6, 73.3 ± 11.6 and 66.7 ± 11.6%, which were not significantly different until the end of the experiments (*p* > 0.05).

## 4. Discussion

In this study, cDNA encoding the hepcidin gene of Nile tilapia (*On*-Hep1) was successfully characterized. The full-length cDNA sequence encoding *On*-Hep1 contains 90 and 70 amino acids for immature and propeptide molecules, respectively, similar to the case for hepcidin molecules found in other vertebrates [15,17]. However, the *On*-Hep1 mature peptide consisted of 26 amino acids adjacent to the motif “RQ/H/KR/H” upstream, which is highly conserved in teleosts or cichlid hepcidins [27]. Additionally, the mature peptide, as a crucial part that binds the bacterial cell wall, is cysteine-rich (6–9 residues) [28], and essentially forms four disulfide bridges, which are important for antimicrobial activity. Eight cysteines were observed in the *On*-Hep1 mature peptide. The theoretical isoelectric point (p*I*) value of the *On*-Hep1 mature peptide is 8.78, similar to those of previously reported hepcidins in other teleosts, which have net cationicity for facilitated interaction with negatively charged microbial surfaces [17].

Previous bioinformatic research has identified two classes of hepcidin proteins in vertebrates, hepcidin antimicrobial peptides 1 (HAMP1) and hepcidin antimicrobial peptides 2 (HAMP2), based on their expression patterns and functional roles [29], suggesting that at least two different hepcidin genes may appear in most vertebrate genomes. The phylogenetic analysis of the hepcidin molecules from different species in our research clearly showed four different clades. *On*-Hep1 and *On*-Hep2 were strongly clustered in the first and second minor groups of the first clade, respectively. Grouper hepcidins 1 and 2 were grouped as the third additional minor group of clade 1. Interestingly, Atlantic salmon, Atlantic cod, zebrafish, blue catfish, channel catfish and higher vertebrate hepcidins were found to separately group for clades 2 and 3, respectively, suggesting that multiple evolutionary events may occur in the teleost hepcidin lineage.

Tissue distribution analysis of the *On*-Hep1 gene in a normal fish showed ubiquitous expression in all tested tissues, with the highest expression in the liver, and moderate expression in the spleen, head kidney, heart and PBLs. These results were similar to those of previous studies of HAMP1 in vertebrates, which were mainly expressed in the liver and moderately in other tissues, while HAMP2 were expressed in only the liver, suggesting that HAMP1 might play a crucial role in innate immune defense in fish in different manners [17,18,30,31,32,33,34].

The expression of *On*-Hep1 after challenge with *S. agalactiae* and *F. columnare* in three tissues, the liver, spleen and head kidney, was clearly demonstrated. The challenge with the highest dose of *S. agalactiae* found that *On*-Hep1 expression concisely reached its peak at 12 h after challenge in all three organs. However, *F. columnare* challenge showed that *On*-Hep1 reached its highest peak at 6 h. This suggests that *On*-Hep1 is an acute-phase protein and prone to greater sensitivity in response to *S. agalactiae* than *F. columnare*. This finding reflects the fact that *S. agalactiae* is more systemic in inducing immune defenses than *F. columnare*, which prefers one to externally colonize the external surface areas of the host, such as the gills and skin, during the early stage of infection [35]. The results of this study were similar to those of Neves et al. (2015) [35], who found that the expression of two hepcidins (Hamp1 and Hamp2) of *Dicentrarchus labrax* infected with 1 × 10^5^ CFU of *P. damselae* spp. *piscicida* PP3 was strongly upregulated in the liver, spleen and head kidney, at days 1 and 2 after injection. These reports mostly supported many studies conducted in various fish species challenged with different pathogens [36,37,38,39]. However, downregulation of hepcidin expression after infection was also found in some fish species [40,41]. In various animal models, diverse inflammatory and infectious stimuli, especially *Streptococcus* species and lipopolysaccharide, strongly induce hepcidin expression in the liver via the induction of IL-6, resulting in a rapid reduction in iron levels in serum [41,42,43,44,45]. These results suggest that various fish pathogens may affect the expression responses of different types of fish hepcidins with different host-pathogen interactions.

To determine the efficacy of rPro*On*-Hep1, sMat*On*-Hep1 and sMat*On*-Hep2 in controlling pathogenic bacteria, MIC and MBC assays were employed. However, no peptide inactivated those target bacteria, even though a maximal concentration of 128 mg/mL was used. A very mild inhibitory effect was observed in only rPro*On*-Hep1 against V. vulnificus. The results of this study are similar to the results of previous reports by Cai et al. [13], who found that recombinant Pro-*Om*hep1 showed even better results than the synthetic mature peptide *Om*hep1. However, these results appear to differ from those of Gui et al. [17], who found that SA-hepcidin1 and SA-hepcidin1 in fish did not inhibit the growth of Vibrio parahaemolyticus, but could inhibit *Staphylococcus aureus*, *Vibrio anguillarum* and *Vibrio alginolyticus*. Xu et al. [46] reported that the protein encoded by Blhepc of Lenok (*Brachymystax lenok*) strongly exhibited antimicrobial activity against all bacterial strains tested for both Gram-negative and Gram-positive bacteria and fungi, such as *Acinetobacter baumannii*, *Aeromonas salmonicida*, *A. hydrophila*, *Staphylococcus aureus*, *Enterococcus faecium* and *Candida glabrata*, by destroying the cell membranes of these microorganisms. Liu et al. [18] found that rTF-Hep of roughskin sculpin fish could inhibit all eight tested bacteria, showing MICs of 5–80 μg/mL. The rTF-Hep also had a high affinity for polysaccharides on the bacterial surface, including LPS, lipoteichoic acid (LTA) and peptidoglycan (PGN), and was capable of agglutinating most tested bacteria. Based on the current data, the antibacterial effects of hepcidins on pathogenic bacteria may depend on (1) fish species, (2) type, source, isoform or variant and concentration of hepcidin and (3) type of pathogenic bacteria, which may have different invasion mechanisms.

The binding activity of FITC-*On*-Heps with various fish pathogens was first demonstrated in the present study. Moreover, sMat*On*-Hep1 and sMat*On*-Hep2 did not completely and strongly bind to all tested bacteria, since they bound only some portions in some cells. Additionally, sMat*On*-Hep1 and sMat*On*-Hep2 coincidentally bound nonpathogenic *B. pumilus* and adhered to pathogenic bacteria in different manners. Furthermore, sMat*On*-Hep1 showed much stronger binding with *F. columnare* and *S. agalactiae* than sMat*On*-Hep2 did, while sMat*On*-Hep2 bound more strongly to *A. hydrophila*. The study of Sow et al. (2007) [47] clearly demonstrated that hepcidin molecules could strongly bind Mycobacterium tuberculosis cells, resulting in the disruption of their membrane, lysis and finally loss of the cytosol. These two peptides more specifically react with pathogenic than nonpathogenic bacteria, which may differ from each other in cell surface molecules. Clumps were normally observed for these two peptides in all pathogenic bacteria experiments, indicating that all tested pathogenic bacteria may interact with or defend against the reacted peptides.

The efficacy of these peptides in enhancing phagocytosis in vitro was also analyzed. All peptides did not elevate PA; however, sMat*On*-Hep1 significantly increased the PI over that of the control and rPro*On*-Hep1- and sMat*On*-Hep2-treated groups, suggesting that sMat*On*-Hep1 is vital for being more efficient at improving antigen engulfment than at enhancing phagocytosed cell numbers. Functionally, sMat*On*-Hep1 may refold into a structure similar to the native structure, and act more effectively than rPro*On*-Hep1. Srinivasulu et al. [48] reported that recombinant hepcidin from an E. coli expression system is less active than synthetic hepcidin, which is the result of incorrectly refolded structures. These results suggest that the biological activity of recombinant peptides is dependent on their refolded structures.

Very little is known about the localization of hepcidin molecules in immune-related cells. Only Sow et al. (2007) [47] demonstrated the localization of hepcidin in the mouse RAW264.7 macrophage cell line and mouse bone marrow-derived macrophages during stimulation with mycobacteria and IFN-g. These two stimuli synergistically induced high levels of hepcidin mRNA and protein in the target cells, and high amounts of hepcidin molecules were specifically detected in macrophage phagosomes using a rabbit anti-mouse hepcidin antibody. In our study, localization of the ingested synthetic sMat*On*-Hep1 and sMat*On*-Hep2 on PBMCs and head kidney leukocytes was clearly observed in the cytoplasm of very few cells for PBMCs and many more for head kidney-derived cells. In PBMCs, the positive cells clearly exhibited round shapes, such as lymphocytes, which were also small. However, strong reactions of these two peptides were observed in leukocytes from the head kidney, which appeared similar to phagocytic macrophages with several large vacuoles. This suggests that both sMat*On*-Hep1 and sMat*On*-Hep2 function with only some specific lymphocytes in the blood circulation system and mainly work with mature leukocytes, especially macrophages. The mechanisms of hepcidin molecule ingestion in these cells are unknown. However, based on current information, ferroportin is known as an iron exporter that specifically binds hepcidin and presents on the surface of absorptive enterocytes, hepatocytes, placental cells and macrophages [49]. It has been shown that, after hepcidin binding, ferroportin is internalized and degraded, leading to the decreased export of cellular iron. In our study, it might be suggested that hepcidin moves within these phagocytes through specific pathways via ferroportin during the first stage, and is eventually fused and ingested into intracellular vesicles within lysosomes [50], resulting in the localization of hepcidin within cytosol compartments. Further studies are needed to clarify this unknown mechanism.

The effects of sMat*On*-Hep1 on Nile tilapia immune responses were examined by intraperitoneal injection. This study demonstrated the crucial functional roles of sMat*On*-Hep1 in enhancing phagocytosis and lysozyme activity, but not complement activity. The results clearly showed that both 10 µg and 100 µg of sMat*On*-Hep1 did not enhance the PA of Nile tilapia PBMCs, but significantly enhanced the PI compared with that of the control group. These results suggest that sMat*On*-Hep1 possesses a crucial ability to stimulate phagocytes, by enhancing their engulfment efficacy during acute period responses at 12–48 h. These two concentrations also strongly elevated serum lysozyme activity concurrently with the PI during 24–48 h, suggesting that increasing Mat*On*-Hep1 strongly affected not only the engulfment of antigens, but also lysozyme activity, which is important for innate immune responses to invading pathogens in Nile tilapia.

Recently, Zhang et al. 2017 [50] demonstrated that two antimicrobial peptides (AMPs) (cathelicidin and β-defensin) are crucial representative molecules of innate immunity, that act by modulating the functions of macrophages and IgM+ and IgT+ B cells. It was clearly shown that these two molecules could significantly enhance the phagocytic, intracellular bactericidal, and reactive oxygen species activities of trout IgM+ and IgT+ B cells. These results suggest that sMat*On*-Hep1 may possess vital activity to modulate macrophages and lymphocytes in Nile tilapia immune responses, similar to some other AMPs.

Furthermore, sMat*On*-Hep1 additionally enhanced or suppressed the expression of some immune-related genes in different target organs with different patterns. The application of sMat*On*-Hep1 clearly elevated the upregulation of *On*-CC1 and *On*-CC2 in head kidney from 1–24 h, and those of *On*-Hep1 itself or *On*-Hep2 from 1–6 h or 1–12 h, respectively. Almost all immune-related genes were significantly upregulated at h 12 and 48, especially iron transport-related genes, including *On*-Hep1, *On*-Hep2 and *On*-Trans. On the other hand, in the spleen, all immune-related genes were strongly suppressed, except *On*-CXC2 and *On*-Hep2, which maintained their significant expression levels during the experimental period. In agreement with much previous research, hepcidin regularly increased the expression levels of many immune-related genes [51,52,53,54]. Additionally, recombinant CiHep of glass carp significantly increased the expression of iron regulator genes (*hepcidins*, *ferroportin and ferritin*), *cytokines* (*TNF-a*, *IL-1b* and *IL-8*) and specific immune responsive genes (*IgM*, *IgD* and *MHC II*) [55]. This suggests that sMat*On*-Hep1 may be a strong regulator that is a vital part of inducing these key immune responsive genes (like other vertebrate hepcidins), which normally function in both innate and acquired immunity.

The most important ability of *On*-Heps, enhancing resistance against *S. agalactiae*, was clearly implicated. Survival analyses of these *On*-Heps demonstrated that all of them are truly effective at protecting Nile tilapia from this harmful pathogen. The greatest protection was observed following the sMat*On*-Hep1 application. The mechanisms by which protection is conferred by sMat*On*-Hep1 could be described and supported by its ability to improve the antigen engulfment efficiency and innate immune responses, especially lysozyme activity, binding to pathogens and upregulation of important immune-related genes.

Previously, it was shown that continuously high levels of hepcidin strongly decreased serum iron levels, presumably reducing bacterial growth [56], and low iron levels also restricted bacterial virulence [57], survival and physiology [58]. *Streptococcus* species have developed various mechanisms to uptake iron from environments with limited available iron. They can directly extract iron from host iron-containing proteins, such as ferritin, transferrin, lactoferrin and hemoproteins, or indirectly extract it by relying on specialized secreted hemophores (heme chelators) and small siderophore molecules (high-affinity ferric chelators) [59]. Accordingly, the acquisition of iron and other metal ions, such as magnesium, calcium and zinc, is crucial for *S. agalactiae* homeostasis [60]. In our study, sMat*On*-Hep1 reduced the Fe and Cu concentrations in the liver and blood corpuscles of treated fish, but not in serum. This finding suggests that the target peptide clearly functions to modulate metals other than Fe and that the highest concentration of this peptide is not enough to affect the reduction in Fe levels in serum.

These results are similar to the results of previous reports by Alvarez et al. [5], who found that hepcidin had a strongly protective effect in sea bass (*Dicentrarchus labrax*) against *V. anguillarum*. In addition, Wei et al. (2018) [55] reported that recombinant CiHep improved the survival rate of *Ctenopharyngodon idellus* challenged with *F. columnare*. The fish that were fed a diet containing recombinant CiHep had a higher survival rate than other groups. The study also showed that recombinant CiHep effectively regulated iron metabolism, causing iron redistribution and decreasing the levels of serum iron at the early stage of infection, indicating that the host responded by withholding iron from pathogens to slow down their rapid growth. Information from Michels et al. [56] strongly supported that continuously high levels of hepcidin lead to a decrease in serum iron levels, presumably reducing bacterial growth.

Basically, hepcidin is enormously enhanced during pathogenic infections such as fungal, viral and bacterial infections [57,61]. Low serum iron levels are normally regulated by hepcidin, and this mechanism acts as a host defense mechanism of vertebrates, specifically developed to limit iron intake for the growth and development of pathogens [62]. Recently, to better understand the mechanisms of hepcidin in increasing immunity in teleosts, Wei et al. (2018) [55] studied the effects of GST-CiHep protein production on grass carp defense mechanisms and synthesized the CiHep peptide, finding that the expression of hepcidin and ferritin was significantly increased by recombinant CiHep activation. Moreover, they also discovered that hepcidins play crucial roles in regulating iron metabolism by transferring dietary, recycled, and stored iron, resulting in increased effective protection of fish against *F. columnare*. This could be the same function as that shown by the experiments in Nile tilapia in this study.

This information indicates that application of sMat*On*-Hep1 could decrease levels of Fe or other metals in fish tissues, to not only inhibit bacterial growth, but also reduce the virulence of bacteria, increasing fish resistance to invading pathogens. To our knowledge, this is the first report of in vivo hepcidin application that provides crucial information for application in streptococcosis resistance in Nile tilapia. Appropriate field trials are needed to prove this indispensable ability of Mat*On*-Hep1 in Nile tilapia aquaculture.

In the present study, *On*-Heps were successfully characterized for their structure and evolutionary relationships at the molecular level and compared with various hepcidins reported in vertebrates. qRT-PCR analyses clearly implied significant responses of *On*-Hep1 to two severely pathogenic bacteria, *S. agalactiae* and *F. columnare*, as an acute-phase factor. The results also clearly showed that rPro*On*-Hep1, sMat*On*-Hep1 and sMat*On*-Hep2 were effective against pathogenic bacteria, especially sMat*On*-Hep1, even though the in vitro antimicrobial activity was unclear. sMat*On*-Hep1 was obviously detected in lymphocyte-like cells from PBMCs and macrophage-like cells isolated from the head kidney. Additionally, it crucially functions as an immunoregulator important for elevating immune responses, especially phagocytic and lysozyme activities, the expression of crucial immune-related genes and the regulation of iron ion levels in the liver. Importantly, among all tested peptides, sMat*On*-Hep1 perfectly proved its ability to protect Nile tilapia from streptococcosis. Information obtained from this study is important for properly implementing an effective strategy to prevent harmful diseases caused by *S. agalactiae* in the Nile tilapia aquaculture industry.

## Figures and Tables

**Figure 1 biomolecules-10-01132-f001:**
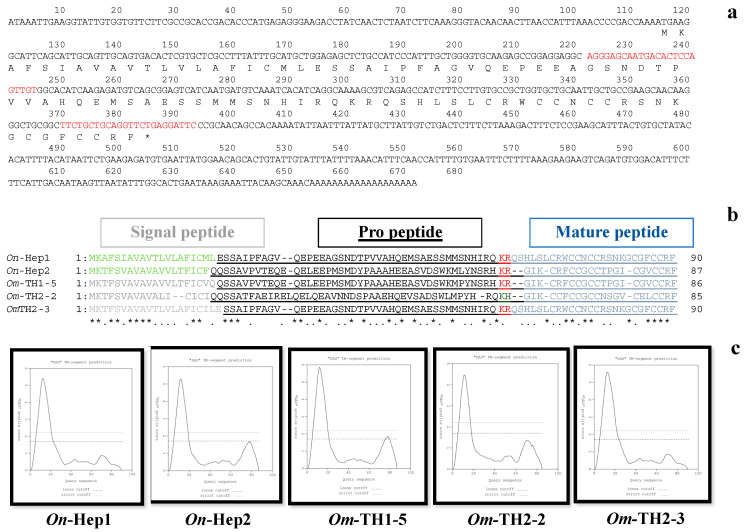
Nucleotide and amino acid sequence analyses of *On*-Hep1. Nucleotide and amino acid sequences of full-length cDNA of *On*-Hep1 (**a**). Comparison of the amino acid sequence structures of hepcidin proteins in Cichlid (**b**). Transmembrane domain prediction of hepcidin proteins in Nile tilapia (*On*-Hep1 and *On*-Hep2) and Mozambique tilapia (*Om*-TH1-5, *Om*-TH2-2 and *Om*-TH2-3) (**c**).

**Figure 2 biomolecules-10-01132-f002:**
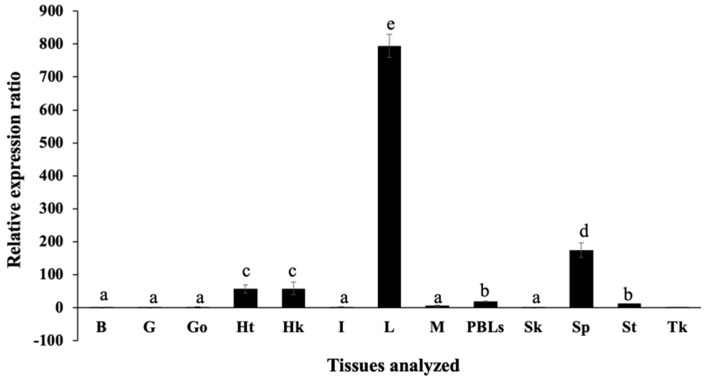
Expression analysis of *On*-Hep1 transcripts using qRT-PCR in 13 tissues. B = brain; G = gills; Go = gonad; Ht = heart; Hk = head kidney; I = intestine; L = liver; M = muscle; PBLs = peripheral blood leukocytes; Sk = skin; Sp = spleen; St = stomach and Tk = trunk kidney. Values of each bar are represented by the means ± SD. Different superscripts on each bar indicate significant differences (*p* < 0.05), n = 3.

**Figure 3 biomolecules-10-01132-f003:**
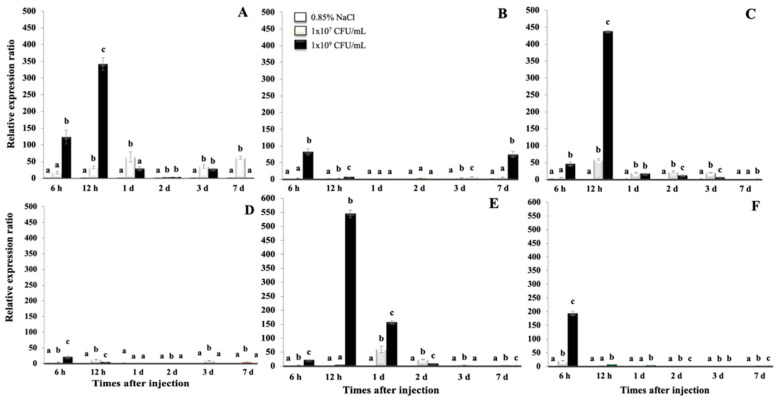
Expression analysis of *On*-Hep1 transcripts in the liver (**A**,**B**), spleen (**C**,**D**) and head kidney (**E**,**F**) of Nile tilapia, in response to *Streptococcus agalactiae* and *Flavobacterium columnare*, respectively. Different superscripts on each bar indicate significant differences (*p* < 0.05), n = 3.

**Figure 4 biomolecules-10-01132-f004:**
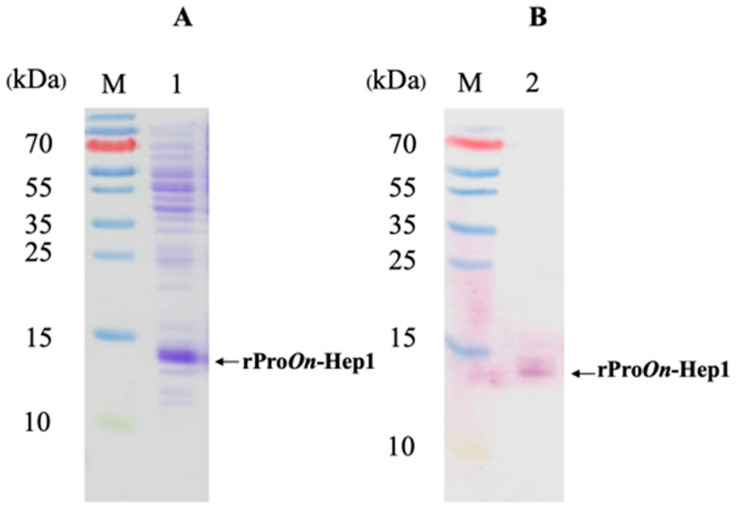
Western blot analysis of rPro*On*-Hep1. SDS-PAGE showing the production of Pro*On*-Hep1 protein. Lane M: protein marker, 1: Pro*On*-Hep1 in cell lysis product (**A**). Western blot analysis of Pro*On*-Hep1 protein. Lane M: protein marker, 2: hybridized Pro*On*-Hep1 with anti-6X His monoclonal antibody (**B**).

**Figure 5 biomolecules-10-01132-f005:**
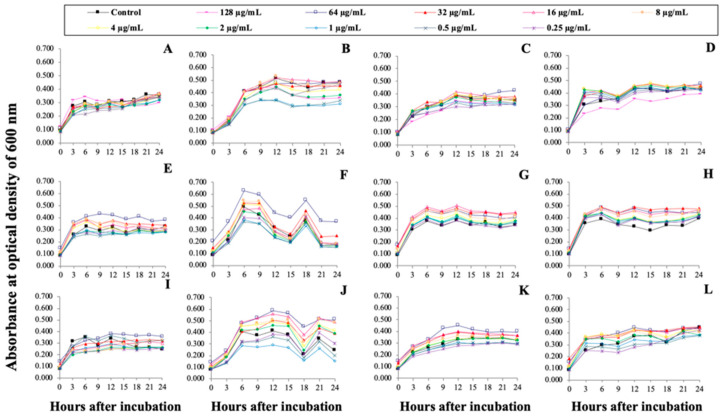
Antibacterial analyses of rPro*On*-Hep1 (**A**)–(**D**), sMat*On*-Hep1 (**E**)–(**H**) and sMat*On*-Hep2 (**I**)–(**L**) abilities to control pathogenic bacteria. Results with *A. hydrophila* (**A**,**E**,**I**), *S. agalactiae* (**B**,**F**,**J**), *V. parahaemolyticus* (**C**,**G**,**K**) and *V. vulnificus* (**D**,**H**,**L**) are shown. The growth of each bacterium was determined by measuring the absorbance at 600 nm.

**Figure 6 biomolecules-10-01132-f006:**
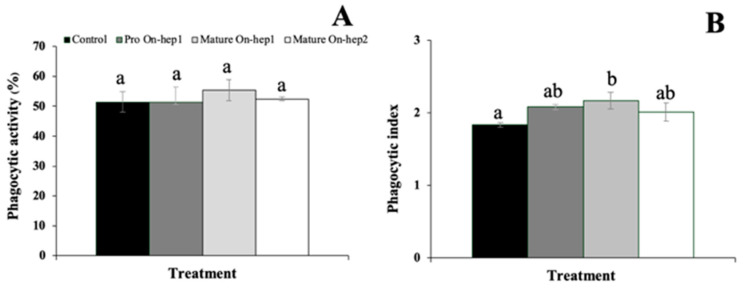
Effects of rPro*On*-Hep1, sMat*On*-Hep1 and sMat*On*-Hep2 on phagocytosis, phagocytic activity (**A**) and phagocytic index (**B**). Different superscripts on each bar indicate significant differences (*p* < 0.05).

**Figure 7 biomolecules-10-01132-f007:**
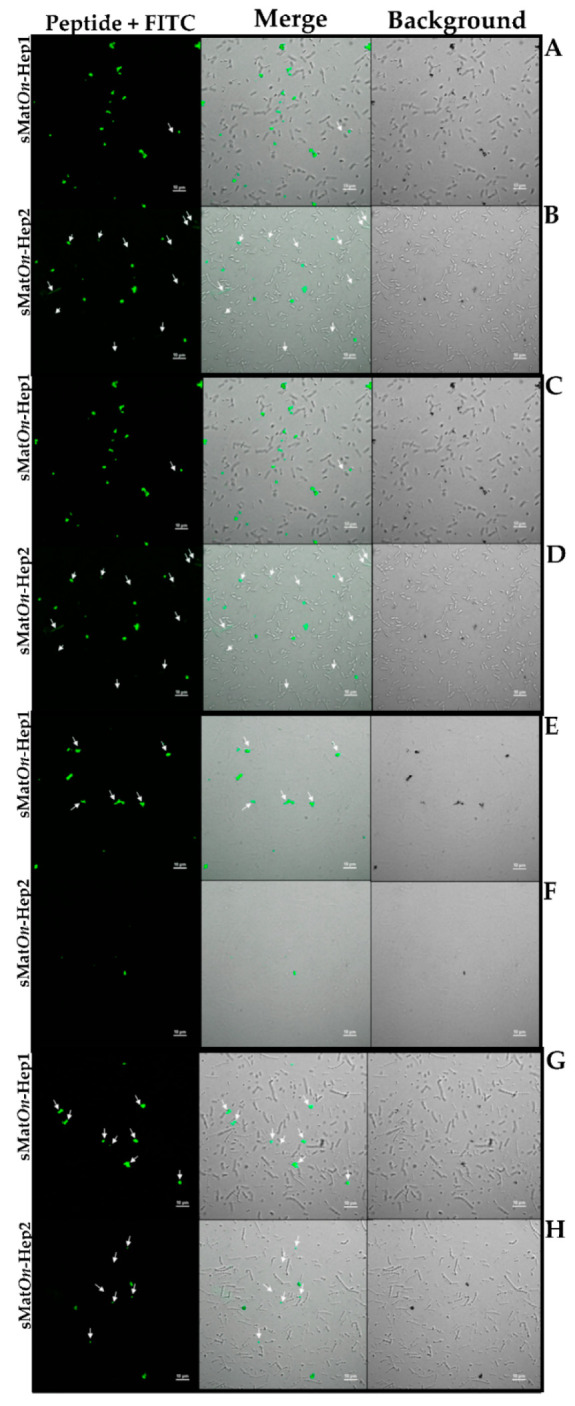
Binding of sMat*On*-Hep1 and sMat*On*-Hep2 to pathogenic bacterial cells, *A. hydrophila* (**A**,**B**), *B. pumilus*, a nonpathogenic control (**C**,**D**), *F. columnare* (**E**,**F**) and *S. agalactiae* (**G**,**H**). White arrows indicate cell-*On*-Hep binding areas.

**Figure 8 biomolecules-10-01132-f008:**
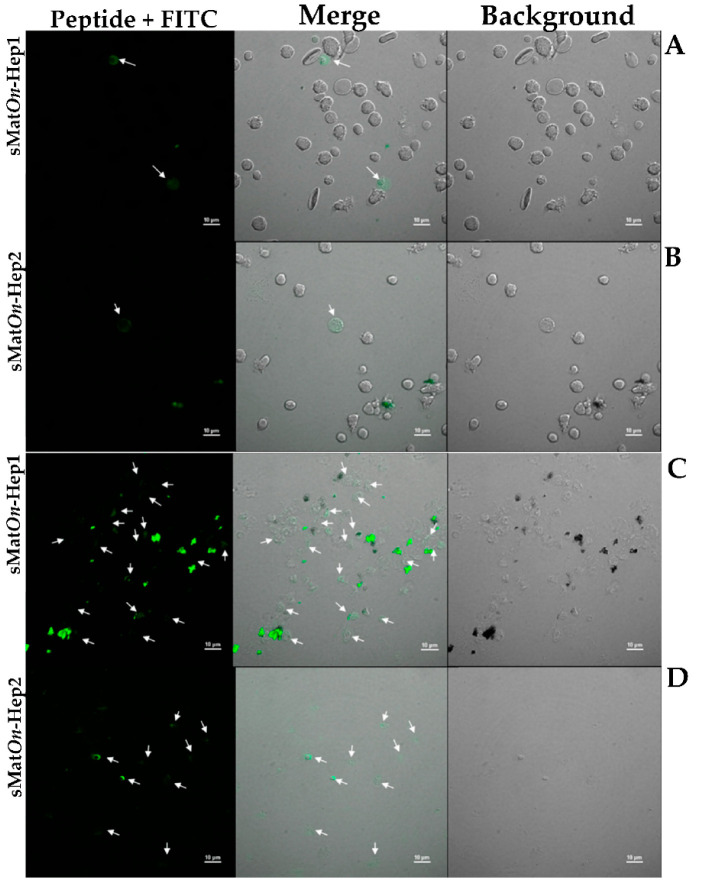
Localization of sMatOn-Hep1 and sMat*On*-Hep2 on PBMCs (**A**,**B**) and leukocytes isolated from head kidney (**C**,**D**). White arrows indicate cell-*On*-Hep binding areas.

**Figure 9 biomolecules-10-01132-f009:**
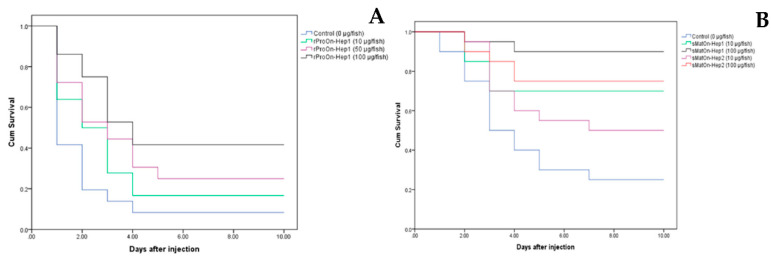
Effects of rPro*On*-Hep1 (**A**), sMat*On*-Hep1 and sMat*On*-Hep2 (**B**) on *S. agalactiae resistance*.

**Figure 10 biomolecules-10-01132-f010:**
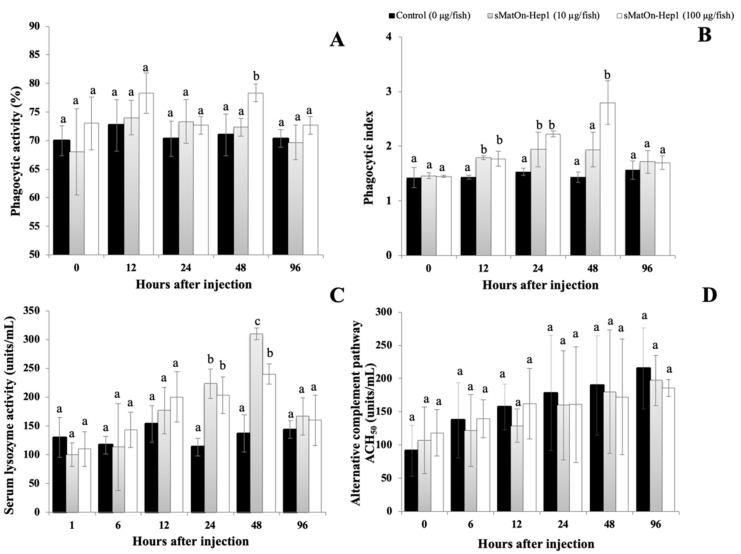
Functional analysis of sMat*On*-Hep1 enhancing innate immune responses. Phagocytic activity (**A**), phagocytic index (**B**), serum lysozyme activity (**C**) and alternative complement pathway (**D**). Different superscripts on each bar at different time points exhibit significant differences (*p* < 0.05), n = 3.

**Figure 11 biomolecules-10-01132-f011:**
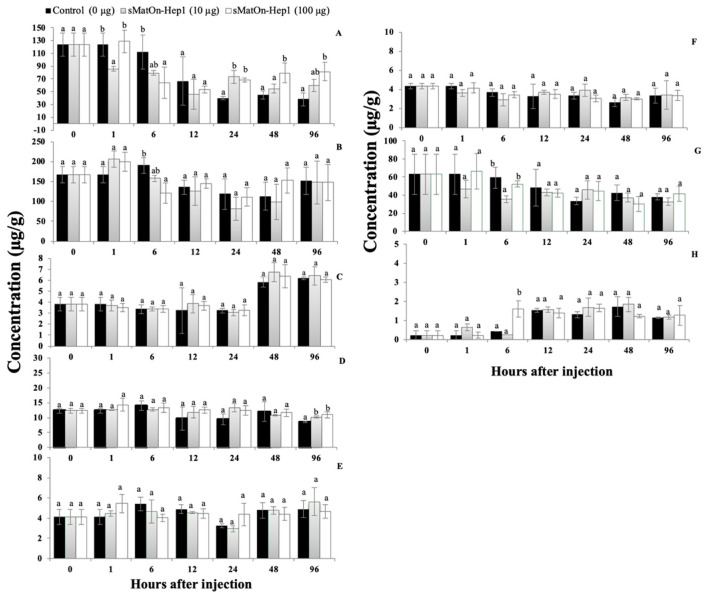
Iron (**A**)–(**C**), zinc (**D**)–(**F**) and copper (**G**)–(**H**) concentrations in the liver, blood corpuscles and serum of Nile tilapia, injected with different concentrations of sMat*On*-Hep1. Different superscripts on each bar at different time points reveal significant differences (*p* < 0.05), n = 3.

**Figure 12 biomolecules-10-01132-f012:**
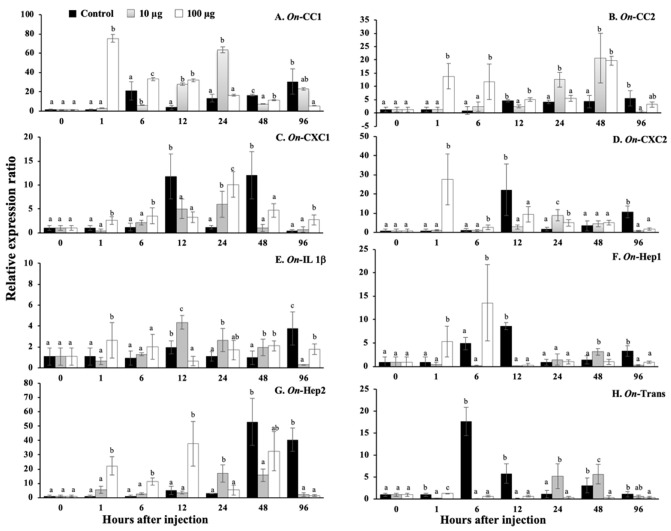
Molecular expression response analyses of the head kidney of Nile tilapia injected with sMat*On*-Hep1 using qRT-PCR. *On*-CC1 (**A**), *On*-CC2 (**B**), *On*-CXC1 (**C**), *On*-CXC2 (**D**), *On*-interleukin-1β (**E**), *On*-hepcidin1 (**F**), *On*-hepcidin2 (**G**) and *On*-transferrin (H). Different superscripts on each bar at different time points indicate significant differences (*p* < 0.05), n = 6.

**Figure 13 biomolecules-10-01132-f013:**
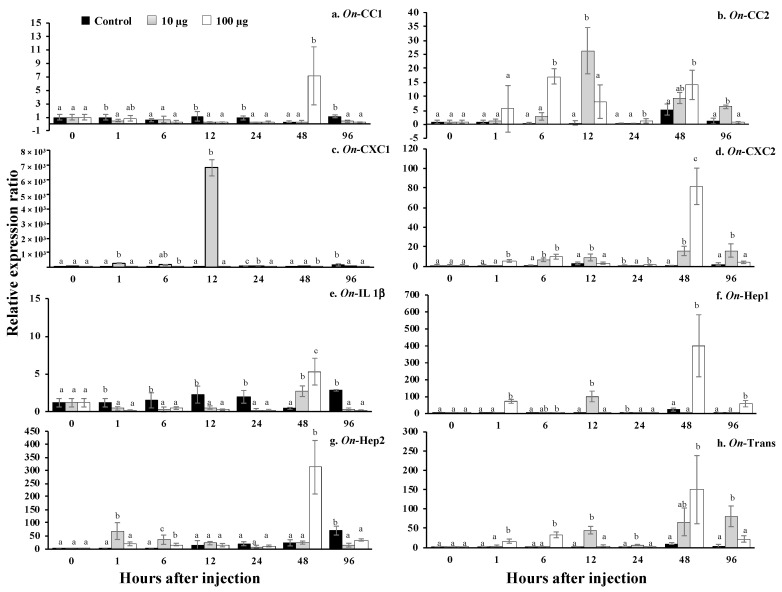
Molecular expression response analyses of the livers of Nile tilapia injected with sMat*On*-Hep1 using qRT-PCR. *On*-CC1 (**A**), *On*-CC2 (**B**), *On*-CXC1 (**C**), *On*-CXC2 (**D**), *On*-interleukin-1β (**E**), *On*-hepcidin1 (**F**), *On*-hepcidin2 (**G**) and *On*-transferrin (**H**). Different superscripts on each bar at different time points indicate significant differences (*p* < 0.05), n = 6.

**Figure 14 biomolecules-10-01132-f014:**
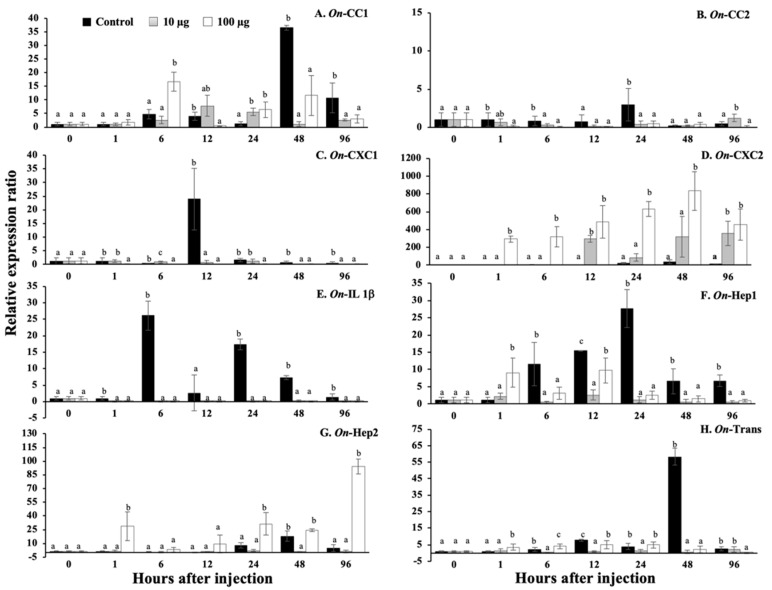
Molecular expression response analyses of the spleen of Nile tilapia injected with sMat*On*-Hep1 using qRT-PCR. *On*-CC1 (**A**), *On*-CC2 (**B**), *On*-CXC1 (**C**), *On*-CXC2 (**D**), *On*-interleukin-1β (**E**), *On*-hepcidin1 (**F**), *On*-hepcidin2 (**G**) and *On*-transferrin (H). Different superscripts on each bar at different time points indicate significant differences (*p* < 0.05), n = 6.

**Figure 15 biomolecules-10-01132-f015:**
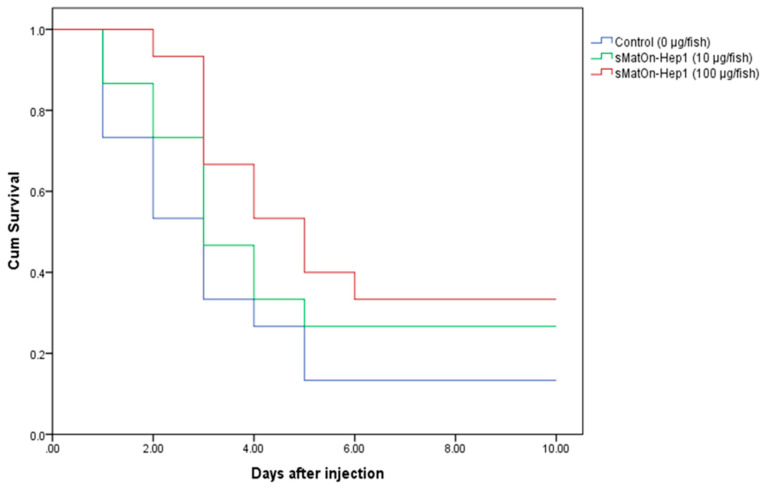
Effects of sMat*On*-Hep1 on *S. agalactiae* resistance after 7 days of application. A survival analysis of the challenge test in each group was performed using the Kaplan–Meier method.

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
