# Peer review of "Immune Regulation, but Not Antibacterial Activity, Is a Crucial Function of Hepcidins in Resistance against Pathogenic Bacteria in Nile Tilapia (Oreochromis niloticus Linn.)"

_biomolecules, 2020, doi:10.3390/biom10081132_

Round 1
Reviewer 1 Report
The present manuscript describes the functions of a recombinant propeptide rProOn-Hep1 and the synthetic FITC-labelled mature peptides sMatOn-Hep1 and sMatOn-Hep2 in Nile tilapia (Oreochromis niloticus Linn.). Based on the results of weak antibacterial function in vitro and the in vivo experiments which can enhance a series of immune functions, the authors propose the point that immune regulation but not antibacterial activity is a crucial function of hepcidins in resistance against pathogenic bacteria in Nile tilapia. Although the rich experimental results are shown in the manuscript to support this conclusion, there are still many problems. First of all, the control group of all the bacterial challenge experiments only injected bacteria, but the experimental group injected both drugs and injected bacteria, this is unreasonable, why the control group did not set the injection PBS? Secondly, there is no further study on whether rProOn-Hep1 with antibacterial activity in vitro have immune functions, so the conclusion of the authors seems uncertain. In addition, I still have some questions and suggestions for the improvement of its quality.
P1: In Figure 4B, the target line of western-blot analysis of ProOn-Hep1 protein is was not clear enough.
P2:In Figure 8, compared to Figure 8C, the leukocytes in Figure 8D are almost invisible, it seems that they are not the same batch of cells.
P3 L617: Format error:1×107; the manuscript does not show why this dose was used as the challenge dose, please list specific references or pre-experimental content.
P4 L618: What is the distinction between h 0,6,12 and days 1?
P5 L690-L691: “V. parahaemolyticus and V. vulnificus” should be “V. parahaemolyticus and V. vulnificus”
P6 L734,L744: Please indicate the specific concentration of the peptide, and check other similar missing in the manuscript.
Author Response
Reviewer 1#
The present manuscript describes the functions of a recombinant propeptide rProOn-Hep1 and the synthetic FITC-labelled mature peptides sMatOn-Hep1 and sMatOn-Hep2 in Nile tilapia (Oreochromis niloticus Linn.). Based on the results of weak antibacterial function in vitro and the in vivo experiments which can enhance a series of immune functions, the authors propose the point that immune regulation but not antibacterial activity is a crucial function of hepcidins in resistance against pathogenic bacteria in Nile tilapia. Although the rich experimental results are shown in the manuscript to support this conclusion, there are still many problems.
First of all, the control group of all the bacterial challenge experiments only injected bacteria, but the experimental group injected both drugs and injected bacteria, this is unreasonable, why the control group did not set the injection PBS?
Response: Thank you so much for your kind comments. In this experiment, we designed to conduct the cohabitat challenge conditions, all fish groups must be reared in the same container. We have a great experience that if we provided PBS control group, fish in this group would be always infected accidentally by transmitting from pathogen injected group. Moderate mortality is always recorded in this condition. In order to avoid this result, therefore, we designed to not include PBS control in our challenge experiment.
Secondly, there is no further study on whether rProOn-Hep1 with antibacterial activity in vitro have immune functions, so the conclusion of the authors seems uncertain.
Response: Thank you so much for your concerns for this point. In our research, we early conducted challenge test to evaluate the efficacy of the rProOn-Hep1, sMatOn-Hep1 and sMatOn-Hep2 and it was shown that the sMatOn-Hep1 provided the best protection among tested peptide, and the rProOn-Hep1 posses the less protection. Therefore, we further mainly designed to focus on only sMatOn-Hep1 and sMatOn-Hep2.
In addition, I still have some questions and suggestions for the improvement of its quality.
P1: In Figure 4B, the target line of Western-blot analysis of ProOn-Hep1 protein is was not clear enough.
Response: Thank you so much for your kind comments to this picture. To increase visibility of this picture, we used the higher resolution one, and enlarged it to make it clearer.
P2:In Figure 8, compared to Figure 8C, the leukocytes in Figure 8D are almost invisible, it seems that they are not the same batch of cells.
Response: Thank you so much for your kind comments to this result. We used the same batch of the cells which isolated from the same fish. However, during picture capturing by the confocal microscope, it was shown that the positive cells in this treatment is less than the other groups and very difficult to focus. Additionally, the positive cells that we took look smaller and were difficult to catch up by our microscope.
P3 L617: Format error:1×107; the manuscript does not show why this dose was used as the challenge dose, please list specific references or pre-experimental content.
Response: Thank you for your eagle-eye observation and then we already corrected this error. For these 2 bacterial concentrations that we used is was selected based on their ability to induce moderate and severe responses from Nile tilapia that we preliminary tested. We put this information in to this section of the manuscript.
P4 L618: What is the distinction between h 0,6,12 and days 1?
Response: Thank you so much for your kind comments. Basically, hepcidin is classified as an acute-phase molecule that may respond and activate other immune components at earlier periods. Therefore, we targeted to prove and demonstrate it activity in these periods.
P5 L690-L691: “V. parahaemolyticus and V. vulnificus” should be “V. parahaemolyticus and V. vulnificus”
Response: Thank you so much for your kind comments. We italicized this information.
P6 L734,L744: Please indicate the specific concentration of the peptide, and check other similar missing in the manuscript.
Response: Thank you so much for your kind comments. We specified the concentration of these peptides in these places in “4.9.2. Localization assay”, while “4.8.2. Cell binding analysis” the concentrations of peptide used were already indicated in “4.8.1. Preparation of pathogenic bacteria”.

Reviewer 2 Report
In this manuscript, the authors identify and characterize a type 1 hepcidin in Nile tilapia, evaluate its expression in several tissues and test its various possible functions and cellular interactions, as well as those of other modified mature peptides. There was clearly a lot of work involved, and a lot of new relevant information is provided and definitely contributes for a better understanding of hepcidin, but there are some problems that should be addressed. One of the biggest is the choice of molecules to test. The authors tested a recombinant Hep1 and FITC modified mature peptides Hep1 and Hep2. However, it would have been important to also test recombinant Hep2 and the unmodified mature peptides, to more clearly support their conclusions, as well as the manuscripts’ title. Without testing those, they can’t say for sure that hepcidin does not have a crucial antimicrobial activity. There are also other points that need addressing.
Results
Section 2.1 – a key feature of OnHep-1 was not put in evidence, in particular the possible iron regulatory sequence QSHLSL, which distinguishes between type 1 hepcidins (more iron related) and type 2 hepcidins (more antimicrobial). Also, please add the accession number for the novel On-Hep1 sequence in the main manuscript.
Figure 2 – PBL should be peripheral blood leukocytes and not lymphocytes, according to the materials and methods.
Section 2.6 – It’s a bit odd that, with so many published papers clearly showing antimicrobial activity for many different fish hepcidins, here we see almost no activity. The authors should have tested the mature peptides by themselves. These results are not very informative, because there are several possible problems: (1) the fact that they use a propeptide and not a mature peptide can clearly interfere in any possible activity, by a number of ways, such as incorrect folding (hepcidin needs the hairpin like structure to be fully functional; the use of a recombinant form assuring that it had the correct folding is described in one of the references of this manuscript, Cai et al.); (2) the propeptide might be more stable and have a longer half-life, but “spontaneous” cleavage and release of the functional mature peptide might be impaired or be too slow, hence not enough to see significant activity; (3) the FITC labelled peptides, are they absolutely sure that the modifications do not interfere with their activity? No, because they have not tested the unmodified mature peptides to compare. In my opinion, I believe they will definitely have antimicrobial activity in their pure, unmodified form.
Section 2.12.1 – I’m not entirely convinced by these results that higher levels of hepcidin lead to increases in phagocytic activity. The fact that the authors are exposing cells to higher levels of a FITC modified molecule makes me wonder if more beads are being phagocyted precisely because of that, not induced by hepcidin, but together with hepcidin. I would like to see the beads/FITC ratio in the cells, if bead phagocytosis increased without an increase in intracellular FITC-hepcidin, then these results would be more convincing.
Section 2.12.4 – The results regarding iron levels in the liver are odd. The authors state that initially, administration of synthetic peptides leads to a decrease in liver iron levels, but then later on the opposite occurs. But there are several things that don’t add up. First, why do levels in control animals change so much? From 0 to 96 hours they go from around 110 to less than half of that (around 45). Why? The injection of PBS should not have an effect that significant (and likely, no effect at all in liver iron levels), something else is causing that. And also, what is the explanation for a decrease in iron levels when Hep1 is administered? If hepcidin blocks ferroportin, iron does not leave the hepatocytes (or enterocytes, or macrophages), so more iron should be accumulated or at least remain the same, not less. Why is iron leaving the liver, where is it going? Is there an increase in haematological parameters (haematocrit, haemoglobin levels)? This needs to be addressed. Also, in line 262, it should be Fig 11A, not 14A.
Line 392 – ug/ml
Line 406 – All true, but the problem here might also reside in non-functional or less functional modified peptides.
Line 430 – Very little is known about the localization of hepcidin in immune cells, and these results don’t really increase that knowledge. In mice studies we are talking about endogenous hepcidin, produced by the cells themselves, especially macrophages, when subjected to different stimuli. Here, the authors are talking and detecting exogenous hepcidin given to the cells. Very different things. Also, the authors suggest that hepcidin is internalized by ways of ferroportin interaction but don’t actual consider direct phagocytosis of a FITC modified molecule, basically a “strange” molecule for the cells, although they do refer that further studies are required.
Line 529 – A single On-Hep1 was characterized.
Materials and methods
Section 4.2 – At this scale, and with these many sequences, neighbour-joining phylogenetic analysis is not recommended, due to lack of robustness. Either maximum-likelihood or Bayesian inference would be better suited.
Line 568 – Ethics committee permit number and date should be included.
Line 573 – Isolation of PBLs should be better described. The provided reference is not enough, since it leads to a paper with no explanation, which in turns refers to another paper, which in turn refers to the actual paper where the methodology is described (by Secombes et al). So the authors should either actually describe the methodology used or reference the original paper.
Line 617 – 1x10^7
Line 691 – Concentrations up to 128 ug/ml were used, not 64. Also, molarity would be more informative than concentration, since the different peptides tested likely have different MW.
Line 708 – 10^6 cells/ml
Lines 750/752 – 150 animals were used, but separated in 3 groups of 48 (144 fish total)
Line 766 – The control group was again injected with more bacteria? The control should have been injected with the same solution that was used to dilute the peptides 1 hour after infection.
Section 4.11.3 – Description of this experiment is very confusing, compared with the description of other experiments. You start with 100 fish, then move 50 fish to another tank, then describe 10 fish in each tank marked with 5 colours (so likely two fish per colour) and then the remaining 10 fish in each tank were controls? Please revise this section for better clarification.
Lines 799/801 – Why were the controls injected with twice as much volume as the experimental animals (200 ul versus 100 ul)? Volumes should be the same.
Line 805 – The authors state that the same parameters were analysed for all fish after injection at 1, 6, 12, 24, 48 and 96h. But in the next section about PBMCs, they were isolated at 0, 12, 24, 48 and 96h. Time points don’t match.
Overall
Please recheck species italicization.
Author Response
Reviewer 2#
In this manuscript, the authors identify and characterize a type 1 hepcidin in Nile tilapia, evaluate its expression in several tissues and test its various possible functions and cellular interactions, as well as those of other modified mature peptides. There was clearly a lot of work involved, and a lot of new relevant information is provided and definitely contributes for a better understanding of hepcidin, but there are some problems that should be addressed. One of the biggest is the choice of molecules to test. The authors tested a recombinant Hep1 and FITC modified mature peptides Hep1 and Hep2. However, it would have been important to also test recombinant Hep2 and the unmodified mature peptides, to more clearly support their conclusions, as well as the manuscripts’ title. Without testing those, they can’t say for sure that hepcidin does not have a crucial antimicrobial activity. There are also other points that need addressing.
Response: Thank you very much for your really great suggestions useful for us to improve the quality of our manuscript.
Results
Section 2.1 – a key feature of OnHep-1 was not put in evidence, in particular the possible iron regulatory sequence QSHLSL, which distinguishes between type 1 hepcidins (more iron related) and type 2 hepcidins (more antimicrobial). Also, please add the accession number for the novel On-Hep1 sequence in the main manuscript.
Response: Thank you so much. We added the possible iron regulatory sequence QSHLSL in this part. By the way, accession number of the OnHep-1 (FF280957) was already indicated in section 4.1.
Figure 2 – PBL should be peripheral blood leukocytes and not lymphocytes, according to the materials and methods.
Response: Thank you so much and we the corrected this information.
Section 2.6 – It’s a bit odd that, with so many published papers clearly showing antimicrobial activity for many different fish hepcidins, here we see almost no activity. The authors should have tested the mature peptides by themselves. These results are not very informative, because there are several possible problems: (1) the fact that they use a propeptide and not a mature peptide can clearly interfere in any possible activity, by a number of ways, such as incorrect folding (hepcidin needs the hairpin like structure to be fully functional; the use of a recombinant form assuring that it had the correct folding is described in one of the references of this manuscript, Cai et al.); (2) the propeptide might be more stable and have a longer half-life, but “spontaneous” cleavage and release of the functional mature peptide might be impaired or be too slow, hence not enough to see significant activity; (3) the FITC labelled peptides, are they absolutely sure that the modifications do not interfere with their activity? No, because they have not tested the unmodified mature peptides to compare. In my opinion, I believe they will definitely have antimicrobial activity in their pure, unmodified form.
Response: Thank you so much for a great comment on this point. We well agreed with these three suggestions. With respect to these concerns, however, there are many reports also described about less activity of hepcidin to inactivate some pathogenic bacteria in vitro with low concentrations 1-128 micrograms that similar to our results. Therefore, higher concentration (>128 micrograms) should be tested. However, in our condition 128 microgram of hepcidin is the maximal concentrations that we can prepare, since concentrations at more than 150-200 microgram of the synthetic peptide are very difficult to dissolve and it can directly interfere the obtained results in this part.
Section 2.12.1 – I’m not entirely convinced by these results that higher levels of hepcidin lead to increases in phagocytic activity. The fact that the authors are exposing cells to higher levels of a FITC modified molecule makes me wonder if more beads are being phagocyted precisely because of that, not induced by hepcidin, but together with hepcidin. I would like to see the beads/FITC ratio in the cells, if bead phagocytosis increased without an increase in intracellular FITC-hepcidin, then these results would be more convincing.
Response: Thank you so much for a great comment on this point. We provided the picture of engulfing FITC-incubated beads as below;
Section 2.12.4 – The results regarding iron levels in the liver are odd. The authors state that initially, administration of synthetic peptides leads to a decrease in liver iron levels, but then later on the opposite occurs. But there are several things that don’t add up. First, why do levels in control animals change so much? From 0 to 96 hours they go from around 110 to less than half of that (around 45). Why? The injection of PBS should not have an effect that significant (and likely, no effect at all in liver iron levels), something else is causing that. And also, what is the explanation for a decrease in iron levels when Hep1 is administered? If hepcidin blocks ferroportin, iron does not leave the hepatocytes (or enterocytes, or macrophages), so more iron should be accumulated or at least remain the same, not less. Why is iron leaving the liver, where is it going? Is there an increase in haematological parameters (haematocrit, haemoglobin levels)? This needs to be addressed. Also, in line 262, it should be Fig 11A, not 14A.
Response: Thank you so much for a great comment on this point. We well agreed with these suggestions. We also surprised at the up levels of iron in the control fish that were injected with PBS. In Nile tilapia, which was tested in aquarium or test tank, we observed that most of them always show some abnormally aggressive behaviors by vigorously biting each other after handling for injection. This activity may directly/indirectly affect to some mechanisms involved in iron regulation. To explain the decrease of iron in the liver, unfortunately we did not conduct hematologic parameters that must be useful for narrate this response.
Line 262. Figure 14A was changed to “Figure 11A”
Line 392 – ug/ml
Response: Thank you so much for this point. We corrected this point in the manuscript.
Line 406 – All true, but the problem here might also reside in non-functional or less functional modified peptides.
Response: Thank you so much for this point.
Line 430 – Very little is known about the localization of hepcidin in immune cells, and these results don’t really increase that knowledge. In mice studies we are talking about endogenous hepcidin, produced by the cells themselves, especially macrophages, when subjected to different stimuli. Here, the authors are talking and detecting exogenous hepcidin given to the cells. Very different things. Also, the authors suggest that hepcidin is internalized by ways of ferroportin interaction but don’t actual consider direct phagocytosis of a FITC modified molecule, basically a “strange” molecule for the cells, although they do refer that further studies are required.
Response: Thank you so much for these comments and we much appreciated to learn this suggestion.
Line 529 – A single On-Hep1 was characterized.
Response: Thank you so much for this point. We corrected this point in the manuscript.
Materials and methods
Section 4.2 – At this scale, and with these many sequences, neighbour-joining phylogenetic analysis is not recommended, due to lack of robustness. Either maximum-likelihood or Bayesian inference would be better suited.
Response: Thank you so much for this point. We really would like to reconstruct this phylogenetic tree in order to create robust information. Unfortunately, one of our co-authors, who is the first author of the manuscript is now affected by Covid-19 around her areas. She is now under quarantine situation and do not have such data in her hand. If we still have some time till she backs from that conditions. We are glad to do this matter for improving our manuscript with this point.
Line 568 – Ethics committee permit number and date should be included.
Response: Thank you so much for this suggestion and we have already provided this information in the manuscript.
Line 573 – Isolation of PBLs should be better described. The provided reference is not enough, since it leads to a paper with no explanation, which in turns refers to another paper, which in turn refers to the actual paper where the methodology is described (by Secombes et al). So the authors should either actually describe the methodology used or reference the original paper.
Response: Thank you so much for this suggestion and we have already provided this information in the manuscript by adding the original reference of Chung, S; C.J. Secombes. Analysis of events occurring within teleost macrophages during the respiratory burst. Comp. Biochem. Physiol. 89B (1988) 539-544.
Line 617 – 1x10^7
Response: Thank you so much for this suggestion and we have already corrected this information in the manuscript.
Line 691 – Concentrations up to 128 ug/ml were used, not 64. Also, molarity would be more informative than concentration, since the different peptides tested likely have different MW.
Response: Thank you so much for this suggestion and we have already corrected this information in the manuscript. For concentrations of the peptide used, we calculated based on their original concentrations. Molarity may affect to the exact concentrations experimented in our study.
Line 708 – 10^6 cells/ml
Response: Thank you so much for this suggestion and we have already corrected this information in the manuscript.
Lines 750/752 – 150 animals were used, but separated in 3 groups of 48 (144 fish total)
Response: Thank you so much for this suggestion and we have already corrected this information in the manuscript.
Line 766 – The control group was again injected with more bacteria? The control should have been injected with the same solution that was used to dilute the peptides 1 hour after infection.
Response: Thank you so much for this suggestion and we have already corrected this information in the manuscript. It was PBS, not bacterial suspension.
Section 4.11.3 – Description of this experiment is very confusing, compared with the description of other experiments. You start with 100 fish, then move 50 fish to another tank, then describe 10 fish in each tank marked with 5 colours (so likely two fish per colour) and then the remaining 10 fish in each tank were controls? Please revise this section for better clarification.
Response: Thank you so much for this suggestion and we have already revised this information in the manuscript.
“One hundred healthy Nile tilapia (100.5±9.3 g) were acclimatized in the same environment in a 1,000-L fiberglass tank as described above with slight modification. Then, 50 fish each were brought into a 500-L fiberglass tank containing 400 L of freshwater”
Response: Since we conduct this experiment with duplicate.
Lines 799/801 – Why were the controls injected with twice as much volume as the experimental animals (200 ul versus 100 ul)? Volumes should be the same.
Response: Thank you so much for this suggestion and we have already revised this information in the manuscript.
Line 805 – The authors state that the same parameters were analysed for all fish after injection at 1, 6, 12, 24, 48 and 96h. But in the next section about PBMCs, they were isolated at 0, 12, 24, 48 and 96h. Time points don’t match.
Response: Thank you so much for this suggestion and we have already revised this information in the manuscript.
Overall
Please recheck species italicization.
Line 708 – 10^6 cells/ml
Response: Thank you so much for this suggestion and we have already checked this information throughout in the manuscript.

Reviewer 3 Report
I was honored to review the manuscript entitled "Immune regulation but not antibacterial activity is a crucial function of hepcidins in resistance against pathogenic bacteria in Nile tilapia (Oreochromis niloticus Linn.)" submitted to Biomolecules.
Taking into account the multiple studies ongoing in this field this type of study is needed. I have only few small remarks that authors should address properly.
I recommend to accept the manuscript after minor revision.
There are only some points to correct:
- please provide clear aim of this study
- please provide the list of abbreviations.
- in results please state: higher or significantly higher, also lower or significantly lower?
- Introduction and Discussion sections needs improvement- please cite doi: 10.3390/molecules25071674 ; 10.1007/s12011-019-1658-1 ; 10.17306/J.AFS.0554.
- In discussion please provide “study strong points” and “study limitation” section.
- please correct typos
I recommend to accept the manuscript after minor revision.
Author Response
Reviewer 3#
I was honored to review the manuscript entitled "Immune regulation but not antibacterial activity is a crucial function of hepcidins in resistance against pathogenic bacteria in Nile tilapia (Oreochromis niloticus Linn.)" submitted to Biomolecules.
Taking into account the multiple studies ongoing in this field this type of study is needed. I have only few small remarks that authors should address properly.
I recommend to accept the manuscript after minor revision.
There are only some points to correct:
- please provide clear aim of this study
Response: Thank you so much for your comments. We have already added our clear aims in the first paragraph of the manuscript.
- please provide the list of abbreviations.
Response: Thank you so much for your suggestion. We will gladly provide this part, if it does not conflict with guideline of the journal.
- in results please state: higher or significantly higher, also lower or significantly lower?
Response: Thank you so much for this suggestion and we have already revised this information in the manuscript.
- Introduction and Discussion sections needs improvement- please cite doi: 10.3390/molecules25071674 ; 10.1007/s12011-019-1658-1 ; 10.17306/J.AFS.0554.
Response: Thank you so much for this suggestion. We have already tried to add this information to our manuscript; however, it was very difficult to make it fit in both Introduction and discussion. But we duly note to include such great information for the next coming manuscript that is similar to this article.
- In discussion please provide “study strong points” and “study limitation” section.
- please correct typos
Response: Thank you so much for your suggestion. We will gladly provide this part, if it does not conflict with guideline of the journal.
I recommend to accept the manuscript after minor revision.
Response: Thank you so much for your kind comments and supports.

Round 2
Reviewer 2 Report
I am pleased with the modifications/corrections made to the manuscript, as well as the justifications for the unfeasibility to introduce some minor changes at this point and would like to thank the authors for the additional work to improve the manuscript. I believe it is now suitable for publication.
Minor correction in line 578 - it should read Secombes and not Sechmbes.